# Mitochondrial fusion regulates proliferation and differentiation in the type II neuroblast lineage in *Drosophila*

**Dnyanesh Dubal**[¤a], **Prachiti Moghe**[¤b☯], **Rahul Kumar Verma**[☯], **Bhavin Uttekar**, **Richa Rikhy**[*]

Biology, Indian Institute of Science Education and Research, Pune, India

☯ These authors contributed equally to this work.
¤a Current address: Department of Clinical Neurosciences and MRC Mitochondrial Biology Unit, University of Cambridge, Cambridge, United Kingdom
¤b Current address: European Molecular Biology Laboratory, Heidelberg, Germany
* richa@iiserpune.ac.in

**Data Availability Statement:** All relevant data are within the manuscript and its Supporting Information files.

## Abstract

Optimal mitochondrial function determined by mitochondrial dynamics, morphology and activity is coupled to stem cell differentiation and organism development. However, the mechanisms of interaction of signaling pathways with mitochondrial morphology and activity are not completely understood. We assessed the role of mitochondrial fusion and fission in the differentiation of neural stem cells called neuroblasts (NB) in the *Drosophila* brain. Depleting mitochondrial inner membrane fusion protein Opa1 and mitochondrial outer membrane fusion protein Marf in the *Drosophila* type II NB lineage led to mitochondrial fragmentation and loss of activity. Opa1 and Marf depletion did not affect the numbers of type II NBs but led to a decrease in differentiated progeny. Opa1 depletion decreased the mature intermediate precursor cells (INPs), ganglion mother cells (GMCs) and neurons by the decreased proliferation of the type II NBs and mature INPs. Marf depletion led to a decrease in neurons by a depletion of proliferation of GMCs. On the contrary, loss of mitochondrial fission protein Drp1 led to mitochondrial clustering but did not show defects in differentiation. Depletion of Drp1 along with Opa1 or Marf also led to mitochondrial clustering and suppressed the loss of mitochondrial activity and defects in proliferation and differentiation in the type II NB lineage. Opa1 depletion led to decreased Notch signaling in the type II NB lineage. Further, Notch signaling depletion via the canonical pathway showed mitochondrial fragmentation and loss of differentiation similar to Opa1 depletion. An increase in Notch signaling showed mitochondrial clustering similar to Drp1 mutants. Further, Drp1 mutant overexpression combined with Notch depletion showed mitochondrial fusion and drove differentiation in the lineage, suggesting that fused mitochondria can influence differentiation in the type II NB lineage. Our results implicate crosstalk between proliferation, Notch signaling, mitochondrial activity and fusion as an essential step in differentiation in the type II NB lineage.

**Funding:** RR received the following funding for the study. This study is funded by the Department of Biotechnology, India (http://dbtindia.gov.in/) by the project code BT/PR18898/MED/122/19/2016. The funders had no role in study design, data collection and analysis, decision to publish, or preparation of the manuscript.

**Competing interests:** The authors have declared that no competing interests exist.

## Author summary

Mitochondrial morphology and function are coupled to stem cell differentiation and organism development. It is of interest to examine the mechanisms of interaction of mitochondrial dynamics with signaling pathways during stem cell differentiation. We have assessed the role of mitochondrial fusion and fission in the differentiation of neural stem cells called neuroblasts (NB) in the *Drosophila* brain. Depleting mitochondrial fusion proteins Opa1 and Marf led to mitochondrial fragmentation, loss of mitochondrial activity and proliferation, thereby causing a decrease in the numbers of differentiated cells in each type II NB lineage. Mutants in mitochondrial fission protein Drp1 led to mitochondrial fusion but did not cause any differentiation defects. Decreased Notch signaling by the canonical pathway led to mitochondrial fragmentation and a decrease in differentiated cells in each type II NB lineage. Expression of Drp1 mutants in type II NB lineages depleted of Opa1 and Marf suppressed their proliferation and differentiation defects. Expression of Drp1 mutant in type II NB lineages depleted of Notch also led to a rescue of differentiated progeny in each lineage. Our results implicate crosstalk between Notch signaling, mitochondrial activity and fusion as important steps for proliferation and differentiation in the type II NB lineage.

## Introduction

Mitochondria are sparse and fragmented in stem cells and are an elaborate network in differentiated cells [1–3]. Stem cells largely depend upon glycolysis as an energy source, whereas differentiated cells produce a large amount of ATP by electron transport chain (ETC) activity [1,4,5]. Fused mitochondrial morphology is associated with high membrane potential, increased ETC activity and high ATP production, while low membrane potential, reduced ETC activity and low ATP production is seen in fragmented mitochondria [1]. Mitochondrial architecture is regulated by a balance of fusion and fission events [6]. Proteins belonging to the family of large GTPases are involved in mitochondrial fusion and fission. Optic atrophy 1 (Opa1, or Opa1-like in *Drosophila*) and Mitofusin or Mitochondrial assembly regulatory factor (Marf in *Drosophila* or Mitofusin, Mfn in mammals) facilitate inner and outer mitochondrial membrane fusion respectively while Dynamin related protein 1 (Drp1) is required for mitochondrial fragmentation [7–10]. A balance of levels and activity of these proteins regulates mitochondrial shape in the cell [6]. Further, Opa1 plays a significant role in regulating cristae organization in addition to inner membrane fusion [11]. Opa1 oligomerization and inner membrane cristae organization is important for ETC activity. The presence of elaborate cristae leads to organization of ETC complexes as super complexes and enhances their activity [12]. Hyperfusion of mitochondria protects them from degradation in autophagy and also loss of ETC activity [13–16].

Recent studies show that alteration of mitochondrial dynamics affects signaling pathways such as the Notch signaling pathway during stem cell differentiation. The Notch receptor is a transmembrane protein activated by ligands such as Delta. The Delta-Notch interaction is followed by cleavage of the Notch intracellular domain (NICD) in the signal receiving cell. NICD enters the nucleus and regulates gene expression along with Suppressor of hairless (Su(H)) by the canonical pathway thereby providing a signal for proliferation or differentiation [17]. Fragmented mitochondrial morphology maintained by Drp1 in ovarian follicle cells in *Drosophila* is crucial for activating Notch signaling [18,19]. Similarly, loss of Opa1 and Mfn leading to mitochondrial fragmentation in mouse embryonic stem cells causes hyperactivation of Notch

and reduces differentiation of ESCs into functional cardiomyocytes due to loss of calcium buffering [20]. On the other hand, activation of Notch signaling by depletion of mitochondrial fusion and increasing reactive oxygen species (ROS) enhances differentiation in mammalian neural stem cells [21]. Thus mitochondrial fragmentation along with elevated calcium and reactive oxygen species increase has been found to be involved in Notch signaling in these contexts. It is of interest to understand whether Notch signaling induces appropriate mitochondrial morphology in differentiation.

The *Drosophila* neural stem cell or neuroblast (NB) differentiation model has been used effectively to identify regulators of steps of differentiation such as stem cell renewal, asymmetric cell division, polarity formation and lineage development. NBs rely on glycolysis and ETC activity for their energy production during differentiation and tumorigenesis [22–24]. Mitochondrial fusion has recently been found to be essential for tumorigenesis [24]. It remains to be studied whether mitochondrial morphology is also regulated to provide appropriate mitochondrial activity for differentiation in NBs. The type I NB lineage in the larval brain consists of 90 NBs marked by the expression of the transcription factors Deadpan (Dpn) and Asense (Ase). These NBs divide asymmetrically to give rise to a ganglion mother cells (GMCs) marked by the expression of Prospero (Pros) in the nucleus [25,26]. NBs of the type II lineage express Dpn and are 8 in number per optic lobe [27]. Like the mammalian neural stem cell differentiation type II NBs undergo multiple steps of differentiation by forming transit amplifying cells called intermediate neural precursor cells (INPs). Newly formed INPs are smaller in size as compared to type II NBs and do not express Ase and Dpn. Some immature INPs express Ase. Immature INPs undergo a defined series of transcriptional changes to form mature INPs. Mature INPs express Dpn and Ase and proliferate to form GMCs that express nuclear Pros and Ase. GMCs in both the type I and type II lineages finally differentiate into neurons or glia (Fig 1A). Young neurons are present at the base of the lineage and express increased nuclear Pros. Elav is expressed in all neurons [28]. Notch signaling regulates number and differentiation in the type II NB lineage [29,30].

In this study, we have assessed the role of mitochondrial morphology proteins Opa1, Marf and Drp1 in regulating type II NB differentiation. RNAi mediated knockdown of mitochondrial fusion proteins Opa1 and Marf led to mitochondrial fragmentation, loss of mitochondrial activity and defects in proliferation and differentiation in the type II NB lineage while NB number and polarity remained unaffected. Opa1 depletion led to decreased proliferation of the type II NBs and mature INPs and decrease in mature INPs, GMCs and neurons in the type II NB lineage. Marf depletion led to decreased proliferation of GMCs and loss of neurons in the type II NB lineage. On the other hand there was no defect in differentiation in NBs overexpressing a mutant form of mitochondrial fission protein Drp1. Inhibition of mitochondrial fragmentation in Opa1 and Marf depletion in combination with Drp1 mutant overexpression suppressed the differentiation defects suggesting that fused mitochondria are essential for differentiation in the type II NB lineage. Further, Notch depletion led to fragmented mitochondria and loss of differentiation. Increased Notch activity showed mitochondrial clustering. Mitochondrial fusion in the type II NB lineage deficient of Notch led to differentiation. Our results show that mitochondrial fusion interacts with Notch signaling to drive differentiation in the type II NB lineage.

## Results

### Depletion of mitochondrial morphology proteins Opa1, Marf and Drp1 leads to change in mitochondrial organization in type II NBs and mature INPs

We depleted Opa1, Marf and Drp1 to investigate the effect of perturbation of mitochondrial morphology on NB numbers and differentiation. We expressed multiple RNAi lines against

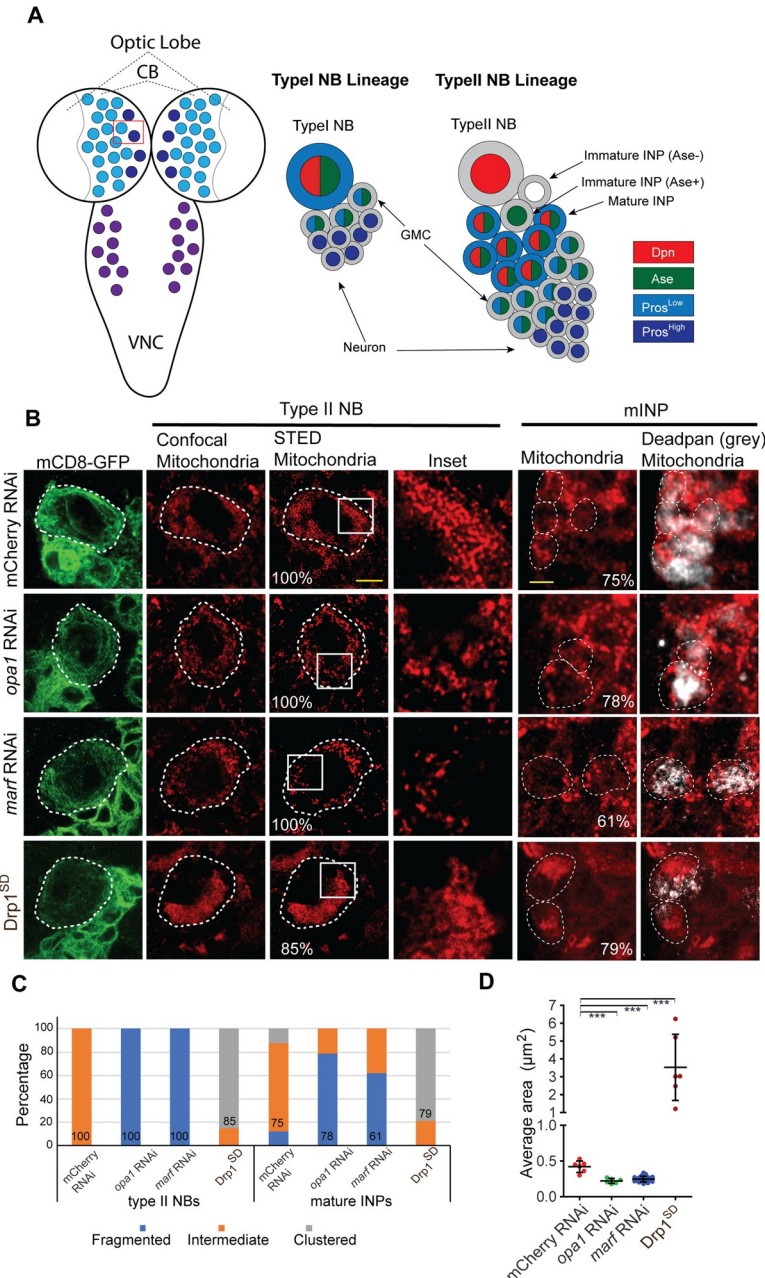

**Fig 1. Opa1 and Marf depletion leads to fragmented mitochondria and overexpression of Drp1<sup>SD</sup> leads to clustered mitochondria in type II NBs.** A: Schematic of larval CNS (left) containing the central brain (CB) lobes and ventral nerve cord (VNC) and type I (blue) and type II NB (purple) distribution and lineages of type I (left) and type II NB (right). The type I NB lineage has nuclear Dpn (red) and Ase (green) and cytoplasmic Pros (light blue) expressing NBs, nuclear Ase (green) and Pros (light blue) expressing GMCs and high levels of nuclear Pros (dark blue) expressing neurons. The type II NB lineage has Dpn positive NBs (red nuclei), Dpn negative, Ase negative and Pros negative immature INPs (black and white), only Ase expressing immature INPs, nuclear Dpn (red) and cytoplasmic Pros (light blue) expressing mature INPs, nuclear Ase and Pros expressing GMCs (green nuclei) and high levels of nuclear Pros (dark blue) expressing young neurons. B-D: Mitochondrial morphology and distribution in type II NBs expressing *pnt*-Gal4, mCD8-GFP (white dotted line, magnified area shown in the panel on the right) stained with ATPβ (red) antibody using confocal imaging and STED super resolution microscopy is shown in representative single optical plane images with zoomed inset in the right panel (B). Single plane confocal images from mature INPs from each image are marked with Dpn and stained with mito-GFP (red). mCherry RNAi (32 type II NBs, 8 brains: 100% intermediate morphology; 40 mature INPs, 5 brains, 8 lineages, 75% intermediate, *opa1* RNAi (46 type II NBs, 8 brains: 100% fragmented; 47 mature INPs, 3 brains, 9 lineages, 78% fragmented), *marf* RNAi (45 type II NBs, 8 brains:

100% fragmented; 21 mature INPs, 3 brains, 4 lineages, 61% fragmented), Drp1$^{SD}$ (80 type II NBs, 10 brains: 85% clustered; 34 mature INPs, 3 brains, 6 lineages, 79% clustered). Graph shows the distribution of mitochondria into fragmented, intermediate and clustered in the form of a stacked histogram (C). The percentage documented on each bar in the histogram is for the group that is seen at the maximum extent. Average mitochondrial area quantification from type II NBs (D) in mCherry RNAi (6 type II NBs, 3 brains), *opa1* RNAi (6,3), *marf* RNAi (19,3), Drp1$^{SD}$(6,3). Graph shows mean ± sd. Statistical analysis is done using an unpaired t-test. *** p<0.001. Scale bar- 5μm for the type II NB in B, 2.7μm for the mature INP in B.

Opa1, Marf and an RNAi line against Drp1 and a dominant negative Drp1 mutation with neuronal Gal4 drivers in different stages of NB differentiation to analyze their effect on lethality and behavior (S1 Fig). *inscuteable*-Gal4, *worniu*-Gal4, *scabrous*-Gal4 and *prospero*-Gal4 were used to deplete Opa1, Marf and Drp1 using RNAi expression and Drp1 using overexpression of a dominant negative mutant in all NBs. Opa1, Marf and Drp1 depletion by multiple RNAi lines and a dominant negative mutant showed survival of animals until the pupal stage with *inscuteable*-Gal4 and *worniu*-Gal4 and were lethal or showed behavioral defects as adults. The RNAi lines for Opa1 and Marf that gave a stronger defect with *inscuteable*-Gal4 and *worniu*-Gal4 and overexpression of the dominant negative mutant of Drp1, (Drp1$^{SD}$) [31] were used to deplete these proteins in the type II NB lineage using *pointedP1*-Gal4, mCD8-GFP (*pnt*-Gal4, mCD8-GFP) for further experiments (Figs 1A and S1). *opa1* RNAi, *marf* RNAi and Drp1$^{SD}$ expression in the type II NB lineage gave normal adults at 25˚C. *opa1* RNAi and *marf* RNAi expression in type II NB lineage when performed at a higher temperature of 29˚C with *pnt*-Gal4 gave sluggish adults.

To characterize mitochondrial morphology in type II NBs, we performed super-resolution Stimulated Emission Depletion microscopy (STED). Mitochondria were stained with an antibody against ATPβ subunit of complex V in the third instar larval brain. STED microscopy allowed better separation of mitochondria in these small cells as compared to confocal microscopy (Figs 1B and S2A). We observed mitochondria in a bead-like organization, often present as spheres evenly distributed all around the nucleus in control type II NBs (Figs 1B and S2A). The mitochondrial morphology was similar to previous observations in type I NBs [32]. We used *pnt*-Gal4, mCD8-GFP to identify the type II NBs using mCD8-GFP and deplete Opa1 and Marf and express the Drp1$^{SD}$ mutant. Pnt expresses in type II NBs, immature INPs and mature INPs [33]. Partial loss of *pnt* leads to a decrease in mature INPs and GMCs in the type II NB lineage. *Pnt*-Gal4, mCD8-GFP shows GFP expression in type II NBs, brighter GFP in cells closer towards the type II NB and lighter GFP in cells towards the base of the lineage. RNAi against mitochondrial fusion proteins Opa1 and Marf has been previously shown to deplete the corresponding mRNA and lead to mitochondrial fragmentation in electron microscopy studies [34–37]. We classified mitochondrial morphology into fragmented, intermediate and clustered in different genotypes in type II NBs and mature INPs by analyzing their distribution in different optical planes (Fig 1B and 1C). We observed an increase in numbers of type II NBs containing mitochondria in a fragmented form often organized as small spheres on depletion of Opa1 and Marf using two different RNAi lines as compared to an intermediate mitochondrial morphology in mCherry RNAi controls, confirming the requirement of these proteins for mitochondrial fusion (Figs 1B, 1C, and S2A). NBs depleted of Opa1 and Marf showed a significant decrease in mitochondrial area as compared to controls (Figs 1D and S2B). The extent of decrease in mitochondrial area was similar upon depletion of either Opa1 or Marf. Mature INPs (Dpn+ cells in the lineage) expressing Opa1 and Marf RNAi also showed increased numbers of cells containing fragmented mitochondria as compared to an intermediate state in controls (Fig 1B and 1C).

Overexpression of mitochondrial fission mutant Drp1$^{SD}$ resulted in clustering of mitochondria on one side of the NB suggesting that mitochondria were fused (Fig 1B). We observed a

similar clustering of mitochondria in mitotic clones of type II NBs depleted of Drp1 using the null allele of *drp1*, *drp1*[KG] [38] (S2C Fig). Mitochondrial clustering on Drp1 depletion has been seen earlier in mammalian cells [39,40] and also in *Drosophila* spermatocytes, neurons, follicle cells and embryos [18,19,31,41,42]. Mitochondrial clusters have inter connected mitochondrial tubes in electron microscopy studies in mammalian cells [40]. Drp1 depletion led to an increase in numbers of type II NBs showing clustered mitochondria and a significant increase in mitochondrial area as compared to control NBs (Fig 1C and 1D). Mature INPs expressing Drp1[SD] also showed an increased number of cells with clustered mitochondria (Fig 1B and 1C). We confirmed the effect of mitochondrial morphology protein depletion on mitochondrial morphology in muscle cells using *mhc*-Gal4, mito-GFP. In line with the previous studies [34,43,44] and NB data, we found that mitochondria were relatively smaller upon depletion of Opa1 and Marf in muscle cells. Expression of Drp1[SD] resulted in relatively large mitochondria compared to control (S2D Fig). Whereas our analysis of mitochondrial distribution could capture changes in mitochondrial morphology at the qualitative level, it could not ascertain changes in mitochondrial size and density of mitochondria in the type II NBs and mature INPs. In summary, depletion of Opa1 and Marf led to mitochondrial fragmentation and depletion of Drp1 led to mitochondrial clustering consistent with mitochondrial fusion in the type II NBs.

## Loss of mitochondrial fusion by depletion of Opa1 and Marf leads to decrease in differentiated cells in the type II NB lineage

We assessed the effect of knockdown of mitochondrial morphology proteins on NB number and differentiation. Depletion of mitochondrial morphology proteins in type II NBs with *pnt*-Gal4, mCD8-GFP did not change their numbers (S3A and S3B Fig). Further we assessed different cell types present in each type II NB lineage based on their specific molecular profile (Figs 1A, 2A and 2B). The numbers of immature INPs (Ase- Pros- cells) were similar in Opa1, Marf and Drp1 depleted type II NB lineages (Fig 2C). Interestingly, mature INPs (Dpn+ Ase+ cells) were significantly decreased in *opa1* RNAi as compared to mCherry RNAi controls (Fig 2A and 2D). Mature INPs give rise to GMCs which further divide to form a pair of neurons. Young neurons contain relatively increased levels of Pros and are located at the base of the lineage. Consistent with the decrease in mature INPs, we observed that GMCs (Dpn- Ase + cells, yellow arrows) (Fig 2A and 2E) and young neurons (Pros high cells) (Fig 2B and 2F) were also reduced in Opa1 depletion. We further confirmed this result of loss of mature INPs and neurons in an independent RNAi line against Opa1 (S3A, S3C, S3D and S3E Fig). Bonnay et al [24] report an increase in mature INPs with a VDRC RNAi line used for depletion of Opa1 in a recent study. We also checked this line for differentiation defects and found that mature INPs and GMCs were decreased on Opa1 depletion (S3F and S3G Fig). We found a consistent decline in mature INPs and GMCs across multiple RNAi lines for Opa1 depletion.

Marf depletion did not alter mature INPs in each type II NB lineage (Figs 2A, 2D, S3A, S3D, S3F and S3G) but led to an increase in GMCs as compared to controls (Fig 2A and 2E). This was also seen in another VDRC line recently used for Marf depletion (S3F and S3G Fig). Conversely, the numbers of young neurons were decreased in *marf* RNAi expressing type II NBs (Figs 2B, 2F, S3C and S3E). The lack of change of mature INPs and increase in GMCs along with a decrease in young neurons in each lineage on Marf depletion implied that GMCs were not able to give rise to neurons.

Depletion of Drp1 causes loss of follicle cell differentiation in *Drosophila* follicle cells [18,19]. However, we did not find any change in numbers of type II NBs and mature INPs, GMCs and young neurons in each Drp1[SD] expressing type II NB lineage as compared to

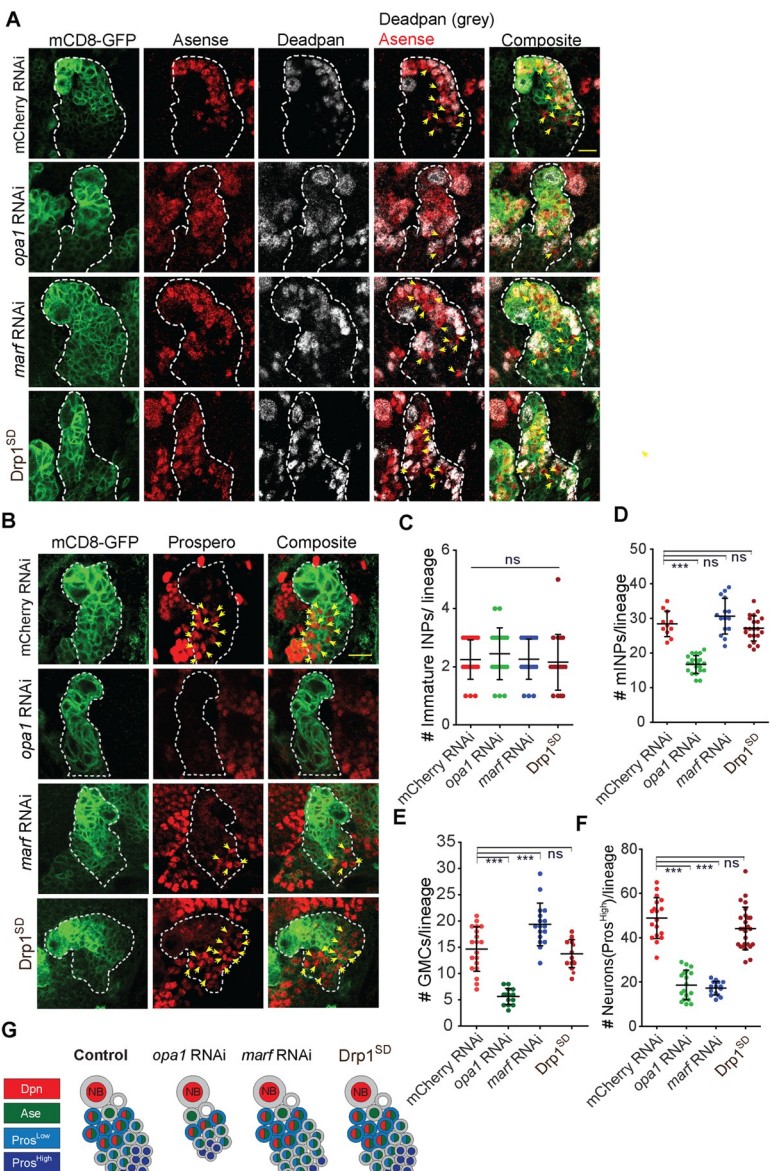

**Fig 2. Analysis of change in mature INPs, GMCs and neurons in the type II NB lineage on depletion of mitochondrial morphology proteins.** A-F: Representative images of the type II NB lineage showing that Dpn positive mature INPs (grey) and Dpn negative, Ase positive GMCs (yellow arrows) are reduced in *opa1* RNAi (A). Quantification of Dpn positive mature INPs per type II NB lineage (D) in mCherry RNAi (n = 11 type II NB lineages, 4 brains), *opa1* RNAi (21,6), *marf* RNAi (15,8), Drp1^SD (28,10). Quantification of GMCs per type II NB lineage (E) in mCherry RNAi (n = 19 type II NB lineages, 6 brains), *opa1* RNAi (14,5), *marf* RNAi (17,6), Drp1^SD (14,5) Scale bar-10μm. Quantification of Ase negative and Pros negative (Ase- Pros-) (C) immature INPs per type II lineage. mCherry RNAi (24 type II NB lineages,5 brains), *opa1* RNAi (20,7), *marf* RNAi (23,8), Drp1^SD (19,8). Representative images of type II NB lineages showing high levels of Pros expressing young neurons are reduced in *opa1* RNAi and *marf* RNAi (B). Quantification of number of young neurons expressing high levels of Pros (yellow arrows point to nuclear Pros) per type II NB lineage (F) of mCherry RNAi (13 type II NB lineages,4 brains), *opa1* RNAi (18,5), *marf* RNAi (19,5), Drp1^SD (26,9). Scale bar- 10μm. G: Schematic depicting the effect of depletions of Opa1, Marf and Drp1 on different cell types in the type II NB lineage. C-F: Graphs show mean ± sd. Statistical analysis is done using an unpaired t-test. ns = non-significant, ***- p<0.001.

controls (Fig 2A, 2B, 2D, 2E and 2F). These data suggest that Drp1 activity is not required for type II NB differentiation while Opa1 and Marf depletions impairs differentiation in the type II NB lineage.

We also depleted mitochondrial morphology proteins using *worniu*-Gal4 to estimate the total numbers of type I and II NBs per brain lobe. The total NB number did not change upon depletion of Opa1, Marf and Drp1 (S4A and S4B Fig). Further we analyzed the apico-basal polarity in NBs by probing polarity markers Bazooka and Numb. Apico-basal distribution of polarity proteins Bazooka and Numb in metaphase is essential for fate determination in the lineage [26]. We found that the NBs did not show a defect in asymmetric localization of Bazooka and Numb on depletion of Opa1, Marf or Drp1 (S4C Fig). However, the numbers of differentiated cells (Pros+ cells) in each brain lobe were decreased in *opa1* RNAi but not *marf* RNAi (S4D and S4E Fig).

In summary, depletion of Opa1 led to a decrease in numbers of differentiated progeny in the type II NB lineage to a greater extent compared to Marf, despite the comparable disruption of mitochondrial morphology. Depletion of mitochondrial fission protein Drp1 did not show any defects in numbers of differentiated progeny in the type II NB lineage (Fig 2G). The phenotype of increased mitochondrial fragmentation and delay in NB division leading to loss of differentiation has been previously noted on depletion of ETC components [23]. Apart from mitochondrial fusion, Opa1 is also involved in the maintenance of cristae architecture [11,45–47]. It is, therefore, possible that a specific defect in mitochondrial inner membrane organization caused by Opa1 depletion in addition to the loss of fusion leads to a decrease in mature INPs and GMCs in the type II NB lineage.

## Drp1$^{SD}$ overexpression along with Opa1 or Marf depletion shows clustered mitochondria and suppresses the defects in differentiation in the type II NB lineage

Mitochondrial fragmentation in Opa1 and Marf depleted cells requires the activity of mitochondrial fission protein Drp1 [31,34,48,49]. To alleviate the mitochondrial fragmentation defect seen in Opa1 and Marf knockdown, we generated combinations of Drp1$^{SD}$;mCherry RNAi, Drp1$^{SD}$;*opa1* RNAi and Drp1$^{SD}$;*marf* RNAi with *pnt*-Gal4 and analyzed mitochondrial morphology in NBs and mature INPs. Unlike *opa1* RNAi and *marf* RNAi (Fig 1B), the Drp1$^{SD}$;*opa1* RNAi and Drp1$^{SD}$;*marf* RNAi combinations showed increased numbers of type II NBs and mature INPs containing clustered mitochondria and these cells also showed mitochondria with an increased area as compared to controls (Fig 3A, 3B and 3C). The mitochondrial cluster in the type II NB was more resolved as compared to controls and Drp1$^{SD}$; mCherry RNAi, with a few punctate mitochondria appearing at the edges of the cluster and a small but significant decrease in mitochondrial area in Drp1$^{SD}$;*opa1* RNAi and Drp1$^{SD}$;*marf* RNAi combinations (Fig 3A and 3C).

We next analyzed the numbers of different cells in the lineage in the Drp1$^{SD}$;*opa1* RNAi and Drp1$^{SD}$;*marf* RNAi combinations (Fig 3D, 3E and 3H). The numbers of mature INPs increased in Drp1$^{SD}$;*opa1* RNAi as compared to *opa1* RNAi and were not significantly different from controls and Drp1$^{SD}$;mCherry RNAi (Fig 3D and 3E). A slight but non-significant increase in the numbers of mature INP in *marf* RNAi was also rescued by co-expression of Drp1$^{SD}$ (Fig 3D and 3E). A similar rescue in GMC (Dpn- Ase+ cells, yellow arrows) numbers were also observed upon co-expression of Drp1$^{SD}$ with both *opa1* RNAi and *marf* RNAi (Fig 3D and 3G). A partial rescue was observed in young neuron (Pros High cells) numbers in Opa1 and Marf depleted type II NB lineages upon forced mitochondrial fusion/clustering (Fig 3F and 3H). Further, we confirmed that rescue in numbers of differentiated cells upon Drp1$^{SD}$

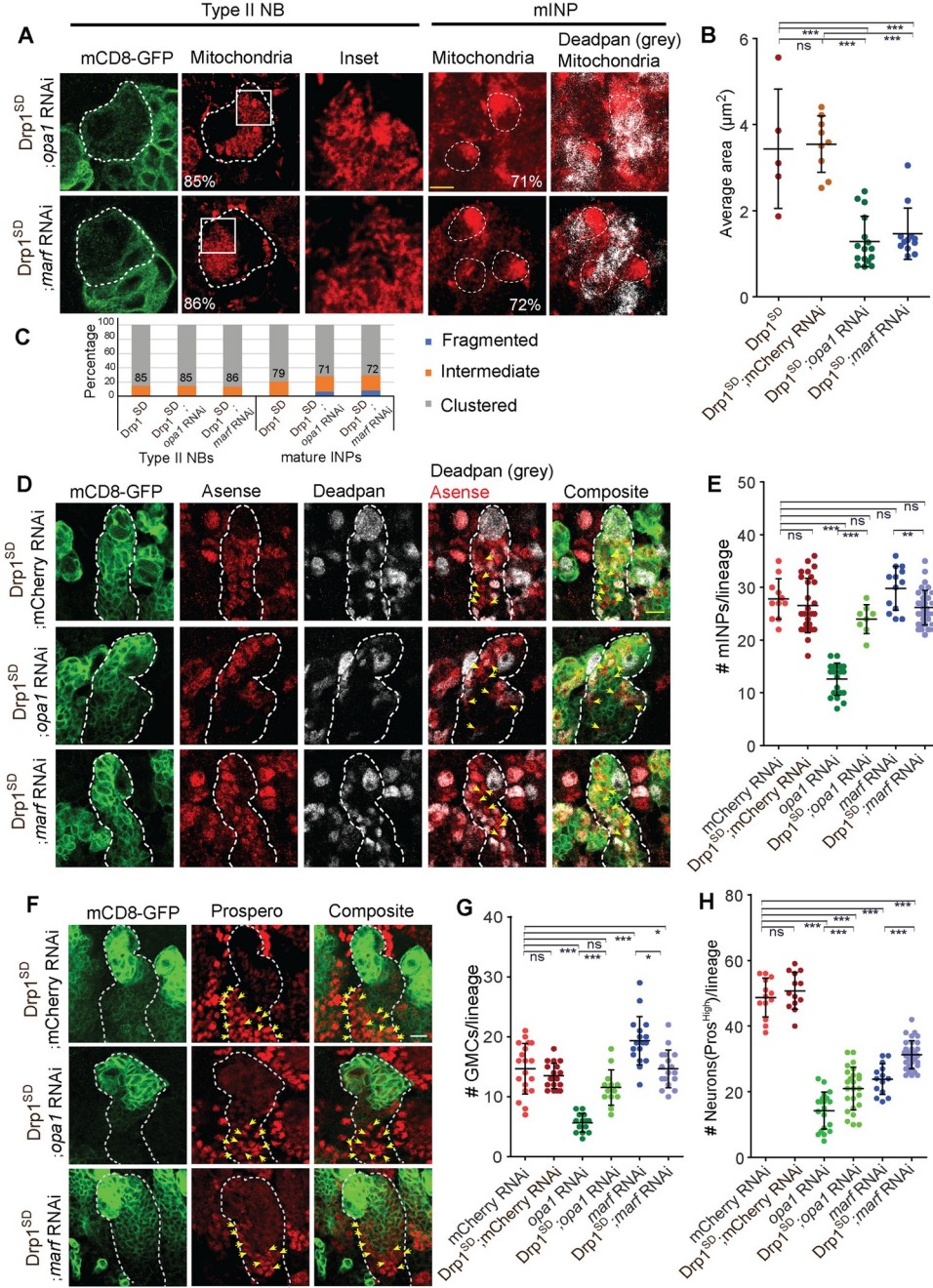

**Fig 3. Drp1$^{SD}$ overexpression in *opa1* RNAi and *marf* RNAi expressing type II NB lineages suppresses the differentiation defect.** A-C: Representative images with zoomed inset in the right panel of type II NBs (white dotted line) containing *pnt*-Gal4, mCD8-GFP (green) stained for mitochondrial morphology with ATPβ (red) and imaged using STED. Mitochondria in mature INPs (Dpn+ cells, grey) are labeled using mito-GFP (red) (A). *pnt*-Gal4, mCD8-GFP with Drp1$^{SD}$;*opa1* RNAi (96 type II NBs,12 brains: 5% clustered; 41 mature INPs, 3 brains, 8 lineages 71% clustered), Drp1$^{SD}$;*marf* RNAi (96 type II NBs,12: 86% clustered; 60 mature INPs, 4 brains, 11 lineages 72% clustered). Quantification of average mitochondrial area (B) in type II NB of mCherry RNAi (8 Type II NBs, 3 brains), Drp1$^{SD}$ (5, 3), Drp1$^{SD}$;mCherry RNAi (9,4),Drp1$^{SD}$;*opa1* RNAi (16,4), Drp1$^{SD}$;*marf* RNAi (12,4). Graph shows the distribution of mitochondria into fragmented, intermediate and clustered in the form of a stacked histogram (C). The percentage documented on each bar in the histogram is for the group that is seen at the maximum extent. The values for Drp1$^{SD}$ are repeated from Fig 1 for comparison. Scale bar- 5μm for the type II NB, 2.7μm for the mature INPs. D,E,G: Type II NB lineages (yellow dotted line) showing expression of mCD8-GFP (green) Ase (red) and Dpn (grey) (D). Quantification of Dpn positive mature INPs (E) in mCherry RNAi (13 NB lineages,4 brains), Drp1$^{SD}$;mCherry RNAi (27,10), *opa1* RNAi (20,8), Drp1$^{SD}$;*opa1* RNAi (8,6), *marf* RNAi (13,8), Drp1$^{SD}$;*marf* RNAi (35,8). Quantification of

Ase positive and Dpn negative GMCs (G, yellow arrows) in mCherry RNAi (19 type II NB lineages,6 brains), Drp1$^{SD}$; mCherry RNAi (18,5), *opa1* RNAi (14,5), Drp1$^{SD}$;*opa1* RNAi (13,5), *marf* RNAi (17,6), Drp1$^{SD}$;*marf* RNAi (17,6). Scale bar- 10μm. F,H: Type II NBs lineages (white dotted line) showing expression of mCD8-GFP (green) and Pros (red, yellow arrows point to nuclear Pros) (F). Quantification of high levels of Pros expressing young neurons (H) in mCherry RNAi (13 NB lineages,4 brains), Drp1$^{SD}$; mCherry RNAi (13,5), *opa1* RNAi (20,8), Drp1$^{SD}$;*opa1* RNAi (25,8), *marf* RNAi (14,8), Drp1$^{SD}$;*marf* RNAi (33,8). Scale bar- 10μm. B,E,G,H: Graphs show mean ± sd. Comparative analysis was done by using unpaired t-test. ns- non significant,*- p<0.1, **- p<0.01, ***- p<0.001.

expression was not due to dilution of the Gal4 by expressing UAS-RFP in Opa1 and Marf depletion backgrounds. Expression of UAS-RFP did not change the altered numbers of Dpn+, Ase+, and Pros high+ cells in type II NB lineages caused by Opa1 and Marf depletions (S5A, S5B, S5C, S5D, S5E and S5F Fig) confirming that rescue in type II NB differentiation was indeed due to reduced activity of Drp1. In summary, the differentiation defect observed on Opa1 and Marf depletion was partially suppressed on mitochondrial fusion with Drp1$^{SD}$.

## Decrease in mitochondrial activity in Opa1 and Marf depleted type II NBs is suppressed by overexpression of Drp1$^{SD}$

Changes in mitochondrial morphology can affect mitochondrial membrane potential and activity. Mitochondrial membrane potential (MMP) is a readout for mitochondrial quality and functionality [50]. We assessed the effect of mitochondrial dynamics proteins depletion on MMP by using the potentiometric dye, Tetra-methyl-rhodamine methyl ester (TMRM) in *vivo* in living larval brains by live imaging. We estimated the relative fluorescence obtained from uptake of TMRM in mCD8-GFP marked type II NBs as compared to neighboring unmarked controls. We found a significant decrease in TMRM fluorescence and therefore MMP on depletion of Opa1 and Marf whereas Drp1$^{SD}$ expression did not affect the MMP (Fig 4A and 4B). Co-depletion of Drp1 along with *opa1* or *marf* RNAi suppressed the MMP defect suggesting that outer mitochondrial membrane fusion restores mitochondrial membrane potential (Fig 4A and 4B).

Decrease in MMP observed in *opa1* and *marf* RNAi may result in a drop in ATP levels in the cell that could trigger a stress response. We have previously found that change in ETC activity but not mitochondrial fusion obtained by Drp1 depletion causes ATP stress in *Drosophila* embryos [31,51]. Depletion of ATP causes an increase in AMP levels which further triggers phosphorylation of AMP activated protein kinase (pAMPK). pAMPK levels elevate in energy deprived conditions and act as an energy sensor inside the cell [52]. To check whether ATP stress was seen in mitochondrial fusion protein depleted NBs, we stained brains with antibodies against pAMPK. NB differentiation relies at least in part on glycolytic metabolism for ATP and loss of ATP in ETC mutants is seen when depleted of both glycolysis and ETC activity [22,23]. Therefore as a positive control for reduction in glycolysis and induction of pAMPK we added 2-deoxy-glucose (2-DG), a non-hydrolysable analogue of glucose to larval brains after dissection and observed a significant increase in pAMPK levels throughout the brain. However, we did not observe a change in pAMPK levels in Opa1 and Marf depleted type II NBs as compared to neighboring NBs indicating that ATP stress similar to 2-DG treatment was not seen in these depletions (S6A and S6B Fig). This shows that activation of pAMPK in NBs depleted of Opa1 and Marf was lower as compared to that seen when glycolysis was inhibited with 2-DG.

Another consequence of change in mitochondrial morphology is alteration in the levels of reactive oxygen species (ROS) [53,54]. We estimated ROS levels using Dihydroxy ethidium (DHE) fluorescence in NBs as compared to neighboring cells. Consistent with previous studies [36,55], we observed an increased DHE fluorescence indicating an increased ROS on Opa1

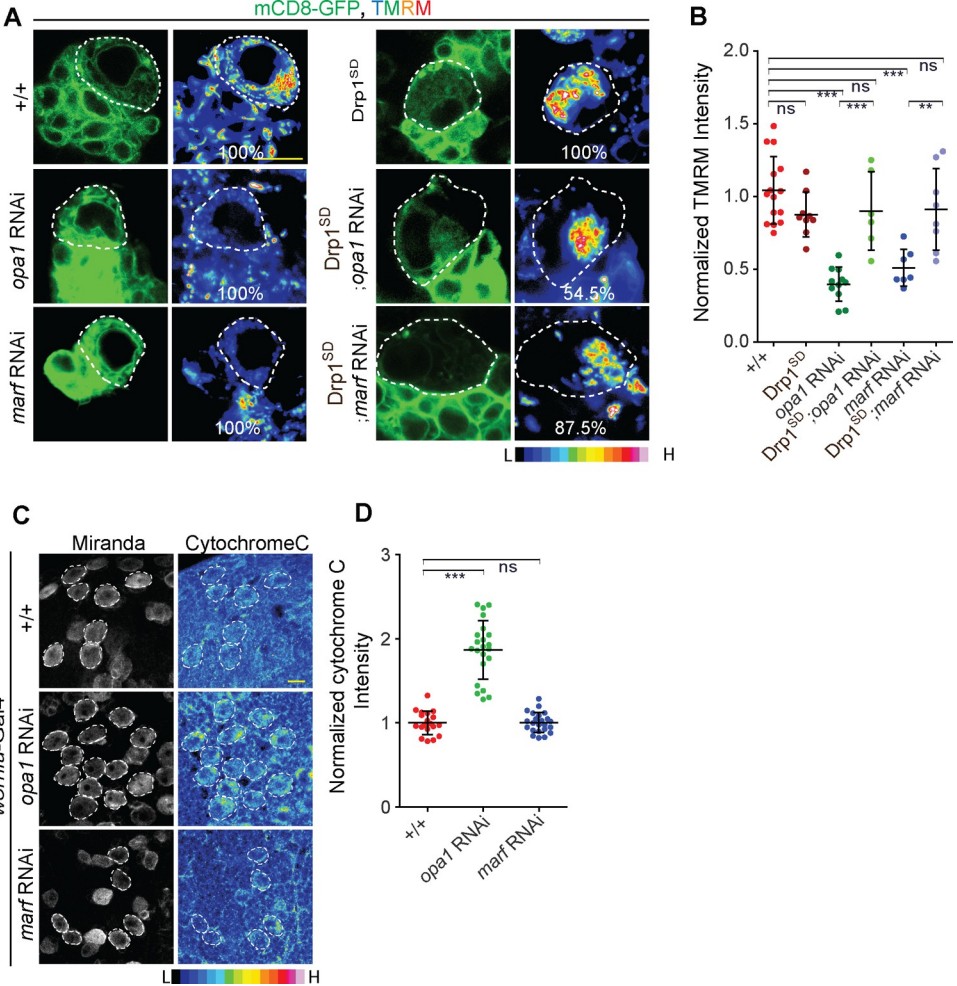

**Fig 4. Drp1^SD overexpression in Opa1 or Marf depleted type II NBs rescues the reduced mitochondrial membrane potential.** A, B: Representative images showing decreased TMRM intensity in *opa1* RNAi and *marf* RNAi expressing type II NBs. Percentage of type II NBs showing similar intensity profile to the representative images is shown on the image panel for each genotype. *pnt*-Gal4, mCD8-GFP (+/+, 17 type II NBs, 8 brains), Drp1^SD (9,5), *opa1* RNAi (16,5), Drp1^SD;*opa1* RNAi (11,4), *marf* RNAi (7,4). Scale bar- 10μm (A). Graph showing normalized TMRM intensities in +/+ (17 type II NBs, 8 brains), Drp1^SD (9,5), *opa1* RNAi (16,5), Drp1^SD;*opa1* RNAi 7,4), *marf* RNAi (11,5), Drp1^SD;*marf* RNAi (6,4) (B). C, D: Representative confocal images of larval NBs (C) showing increased cytochrome C staining in *opa1* RNAi using *worniu*-Gal4. Analysis of cytochrome C intensity normalized with control (D). +/+ (19 NBs, 8 brains), *opa1* RNAi (22 NBs, 7 brains), *marf* RNAi (23 NBs, 7 brains). B,D: Graphs show mean ± sd. Comparative analysis was done by using unpaired t-test. ns- non significant, **- p<0.01, ***- p<0.001.

knockdown and Drp1^SD expression (S7A and S7B Fig). Marf depletion however did not change the DHE fluorescence significantly (S6A and S6B Fig). Since ROS was found to increased in both Opa1 depletion and Drp1^SD NBs, and there was no effect on differentiation on Drp1^SD overexpression (Fig 2), we conclude that ROS is unlikely to play a significant role in differentiation in type II NBs. We further checked if a general increase in ROS can affect development. We found that in type I and II NBs overexpressing a mutant of human Superoxide dismutase (hSOD) [31,56], ROS was increased significantly (S7A and S7B Fig). However, hSOD expression in all NBs did not affect viability of flies and adults emerged similar to controls leading us to conclude that it did not impact differentiation and functionality of neurons.

Even though both Opa1 and Marf depletion led to an equivalent fragmentation and loss of MMP of mitochondria in NBs, Opa1 depletion showed a specific decrease in mature INPs as compared to Marf knockdown. Opa1 oligomerization leads to stabilization of cristae and localization of cytochrome c in cristae. Increase in cytochrome c occurs when Opa1 oligomerization is decreased and this correlates with loose cristae organization [45]. We visualized the distribution of cytochrome c in Opa1 and Marf depleted NBs. Increase in cytochrome c was seen in *opa1* RNAi as compared to *marf* RNAi and controls (Fig 4C and 4D). This suggests that loosening of cristae architecture potentially causes specific spread of the cytochrome c signal on Opa1 depletion. Altogether, loss of MMP in type II NBs and increased cytochrome c in NBs deficient of Opa1 is suggestive of disruption of inner mitochondrial membrane architecture and activity in addition to fusion leading to loss of differentiation.

## Opa1 and Marf depleted type II NB lineages show proliferation defects

The decreased lineage size observed in Opa1 and Marf depleted type II NBs may result from apoptosis or lowered proliferation rates of each NB, thereby reducing the numbers of INPs and GMCs in each lineage. Increased cytochrome C and ROS in Opa1 depletion suggests that type II NB differentiation could be decreased by apoptosis in the lineage [54,57–59]. For probing apoptosis, we assessed the levels of cleaved caspase 3 in *opa1* and *marf* RNAi. Cleaved caspase 3 fluorescence was not elevated in Opa1 and Marf depleted lineages and was similar to controls (S8A Fig). Selectively marking apoptotic nuclei with the Terminal deoxynucleotidyl transferase (TdT) dUTP Nick-End Labeling (TUNEL) assay did not show any significant difference between *opa1* RNAi and control. We validated this assay by using UAS-hid expression as a positive control for apoptosis induction (S8B and S8C Fig). Therefore, cell death is not responsible for the differentiation defect in Opa1 and Marf depleted type II NBs.

Change in mitochondrial morphology impacts the rate of proliferation and differentiation in mammalian stem cells by regulating levels of cyclins [60,61]. Type II NBs divide asymmetrically to produce INPs, mature INPs divide asymmetrically to produce GMCs and GMCs divide to produce neurons and glia in the type II NB lineage. We analyzed the numbers of cells in type II NBs and their lineage cells in the mitotic phase by labeling them with phospho-histone 3 (pH3) antibody. We found that the numbers of type II NBs in mitosis were significantly reduced on Opa1 depletion and this defect was rescued in the Drp1[SD];*opa1* RNAi combination (Fig 5C). The total numbers of pH3 positive cells in the lineage were decreased (Figs 5A and S9C). The total number of cells in G1-S phase as marked by EdU in the lineage were also decreased (Fig 5B and 5F). This decrease in the numbers of cells in the mitotic and G1-S phase was rescued partially on additional expression of the Drp1[SD] mutant (Figs 5A, 5B, 5F and S9C). Drp1[SD] expression alone did not perturb the proliferation of the type II NB or the lineage cells. We also found that there was a decrease in the numbers of mitotic NBs when *opa1* RNAi was expressed with *worniu* Gal4 (S9A and S9B Fig). We further assessed the numbers of mature INPs labeled with Dpn in mitosis in each type II NB lineage with pH3. We found that the mitotic Dpn positive mature INPs were depleted in *opa1* RNAi and this defect was suppressed in Drp1[SD];*opa1* RNAi (Fig 5D). These data together show that decreased division rate of the type II NB gives rise to less mature INPs. The mature INPs produced in turn also divide less thereby impacting GMC and neuron numbers in the lineage.

Depletion of Marf did not impact the division rate of the type II NB when expressed with *pnt* or *worniu* Gal4 (Figs 5C, S9A, and S9B). The division rate of Drp1[SD];*marf* RNAi combinations was slightly increased as compared to *marf* RNAi and controls. There was a significant increase in pH3 and EdU positive cells in each type II NB lineage and this was rescued on additional expression of the Drp1 mutant (Figs 5A, 5B, 5F and S9C). There was a significant

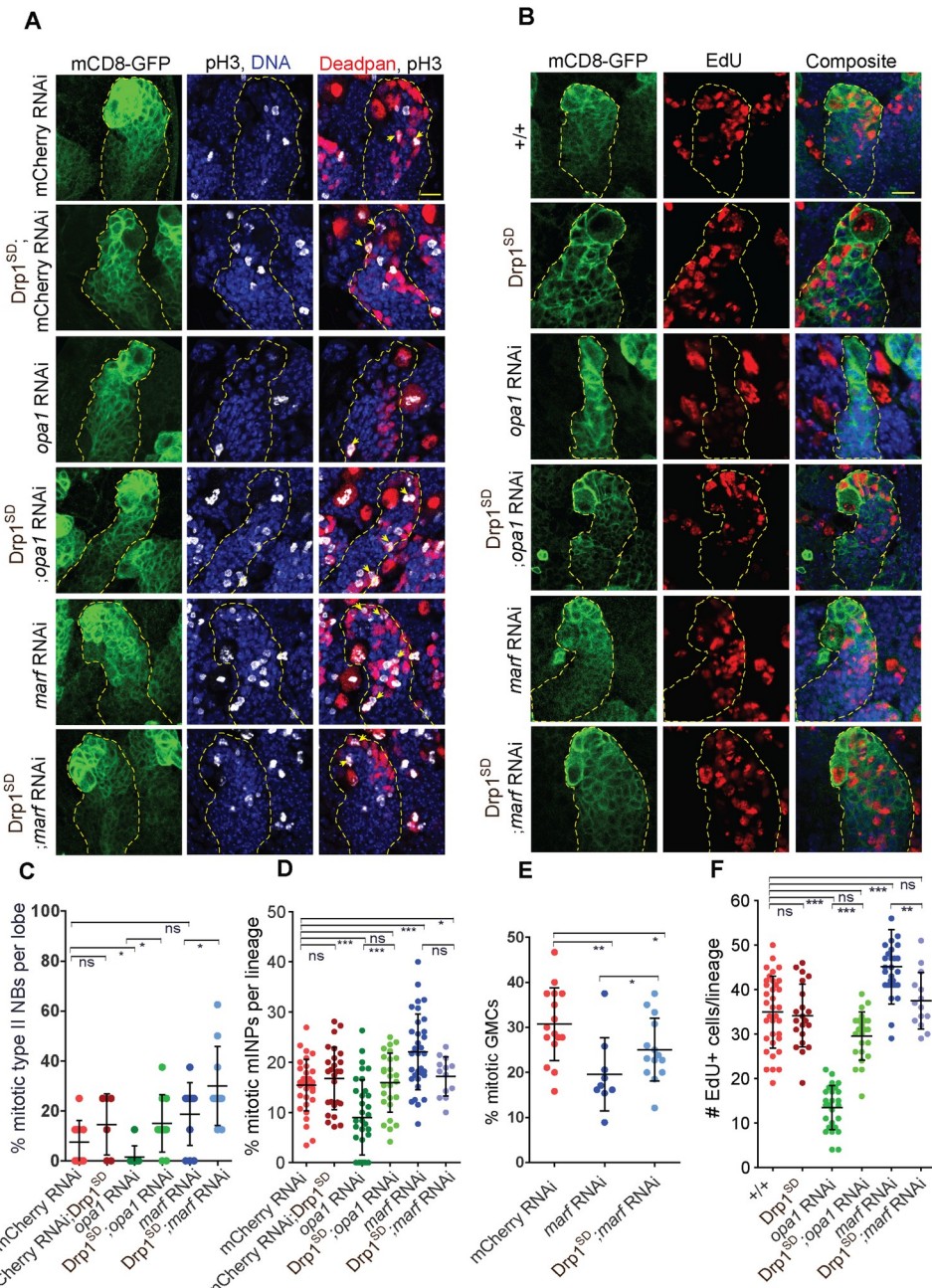

**Fig 5. Proliferation defects in Opa1 depleted type II NB lineages are suppressed by overexpression of Drp1<sup>SD</sup>.** A, C-E: Representative confocal images of type II NB lineages show reduced pH3 positive mature INPs (A) in *opa1* RNAi while increase in *marf* RNAi expressed with *pnt*-Gal4, mCD8-GFP in the type II NB lineage. Analysis of mitotic type II NBs per brain lobe (C) in mCherry RNAi (10 brain lobes), Drp1<sup>SD</sup> (6), *opa1* (8), Drp1<sup>SD</sup>;*opa1* RNAi (10), *marf* RNAi (12), Drp1<sup>SD</sup>;*marf* RNAi (10). Analysis of percentage of dividing mature INPs in type II NB lineages (D, yellow arrows are Dpn+pH3+) in mCherry RNAi (33 Type II NB lineages,7 brains), Drp1<sup>SD</sup>;mCherry RNAi (24,6), *opa1* (33,6, Drp1<sup>SD</sup>;*opa1* RNAi (26,8), *marf* RNAi (33,8), Drp1<sup>SD</sup>;*marf* RNAi (15,5). Quantification of percent dividing GMCs (E) in mCherry RNAi (16 typeII NB lineages, 4 Brains), *marf* RNAi (9,2), Drp1<sup>SD</sup>;*marf* RNAi (14,5), Scale bar- 10μm. Non-parametric unpaired t-test with Welch's correction was used for statistical analysis. ns- non-significant, *- p<0.1, **- p<0.01, ***- p<0.001. B,F:Representative confocal images of type II NB lineages (mCD8-GFP, green) (yellow dotted line) show decreased number of EdU (red) positive cells per lineage (B) in *opa1* RNAi. Quantification of EdU positive cells in type II NB lineages (F) in *pnt*-Gal4, mCD8-GFP +/+ (35 Type II NB lineages, 6 Brains), Drp1<sup>SD</sup> (21,7), *opa1* RNAi (26,6), Drp1<sup>SD</sup>;*opa1* RNAi (23,7), *marf* RNAi (29,7), Drp1<sup>SD</sup>;*marf* RNAi (14,5). Statistical analysis was done by using unpaired t-test.ns- non-significant, **- p<0.01, ***- p<0.001.

increase in mitotic mature INPs in the lineage (Fig 5D). This phenotype was not significantly suppressed by addition of Drp1[SD] even though the variation in numbers of mitotic cells decreased as compared to *marf* RNAi alone. The mitotic GMCs however were significantly decreased and this decrease was suppressed by addition of Drp1[SD] (Figs 5E and S9D). This increase in mitosis in mature INPs and decrease in the mitotic rate of the GMCs causes an increase in GMC numbers and decrease in numbers of neurons.

These data altogether suggest that decrease in the division of type II NBs and mature INPs in *opa1* RNAi and GMCs in *marf* RNAi is responsible for the loss of differentiation in the type II NB lineage.

## Notch signaling drives fused mitochondrial morphology and Opa1 depletion leads to decreased Notch signaling in the type II NB lineage

Notch signaling regulates numbers of NBs and their differentiation in the type II lineage in the *Drosophila* third instar larval brain [29,30,62]. Changes in the cell cycle could impact Notch signaling [63]. Notch signaling is activated in type II NBs and suppressed in immature INPs. Notch signaling is again activated in the mature INPs in the lineage. Loss of Notch signaling leads to decreased type II NBs and transformation of the type II NB to the type I NB with absence of Dpn positive mature INPs. Increased Notch signaling gives rise to increased NBs due to dedifferentiation of immature INPs [30,64]. NBs formed on overexpression of Notch[FL] express Dpn but do not express Ase (S10A and S10B Fig). Also depletion of Opa1 and Marf led to a significant decrease in numbers of NBs in the Notch overexpression background (S10C and S10D Fig).

Since Notch signaling regulates differentiation within each type II NB lineage (Fig 6A and 6B) and loss of mitochondrial fusion proteins led to loss of differentiation in the type II NB lineage, we assessed if Notch signaling regulates mitochondrial morphology in type II NBs. Elongated and fused mitochondrial morphology has been observed in the type II NBs depleted the Notch inhibitor, Numb [24]. We also observed a change in the mitochondrial morphology on increasing Notch activation in type II NBs which were positive for Dpn (Fig 6C). Overexpression of the Notch full length, Notch[FL] and the Notch intracellular domain, Nintra in type II NBs led to a significant increase in cells containing clustered mitochondria on one side of the nucleus (Fig 6C and 6D). Overexpression of Notch[FL] and Nintra led to an increase in the area of mitochondria as compared to control NBs (Fig 6C and 6E). Notch downregulation by expression of *notch* RNAi led to an increase in numbers of cells containing fragmented mitochondria and a decrease in average size of punctae (Fig 6C and 6E). Likewise, depletion of Su(H) in type II NBs also resulted in increase in numbers of cells containing fragmented mitochondria and decreased area of mitochondria (Fig 6C and E). These data show that Notch activation via the canonical pathway through Su(H) leads to mitochondrial clustering and fusion whereas Notch depletion results in fragmented mitochondria. We also assessed mitochondrial distribution on Notch overexpression and downregulation in the mature INPs. We found that mitochondria were clustered on Notch overexpression in greater numbers of mature INPs and they were dispersed on Notch and Su(H) depletion in greater numbers of mature INPs as compared to controls (S10E and S10F Fig). Expression of Notch mutants in muscle cells also gave a similar result of larger mitochondria in Notch overexpression and smaller mitochondria on depletion of Notch in Notch RNAi and Su(H) RNAi (S10G Fig).

Since Opa1 depletion led to decrease in the number of mature INPs, we tested the status of Notch receptor in the type II NBs. We found that the Notch receptor accumulated in the cytoplasm in Opa1 depleted type II NBs as visualized by the Notch intracellular domain antibody (Fig 6F, 6G and 6H). We also observed perturbed localization of Notch receptors in the differentiated cells in the lineage on Opa1 depletion (Fig 6F and 6H). This phenotype was

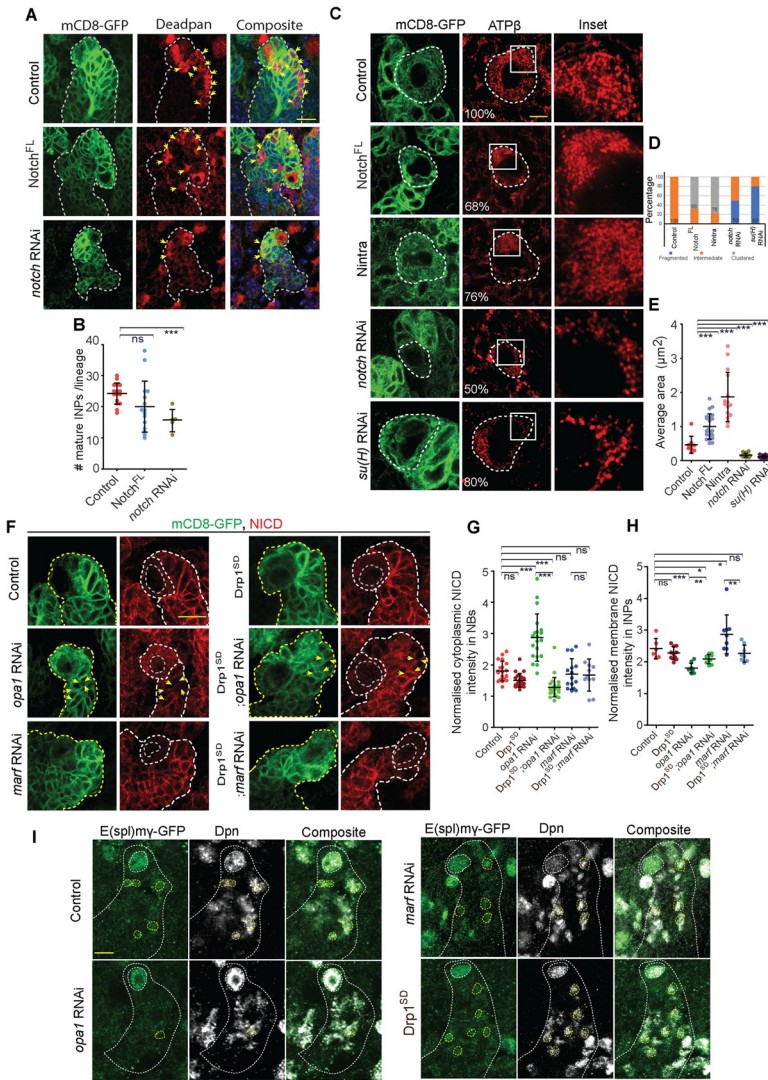

**Fig 6. Analysis of mitochondrial morphology in Notch mutants and Notch signaling in Opa1 depleted type II NB lineages.** A:B: Representative confocal images of type II NB lineages (A) showing reduced numbers of mature INPs in *notch* RNAi. Quantification of mature INPs (B) in Control (16 type II NB lineages, 4 brains), Notch[FL] (15,5), *notch* RNAi (5,3). C-E: Type II NBs (mCD8-GFP, green, yellow dotted line) showing Notch signaling mediated regulation of mitochondrial morphology (red) by ATPβ antibody using STED microscopy (C). +/+ (100% tubular, 75 NBs, 22 Brains), Notch[FL] (68% clustered, 103,16), Nintra (76% clustered, 58,6), *notch* RNAi (50% fragmented, 30,14), *su(H)* RNAi (80% fragmented, 23,12). Graph shows the distribution of mitochondria into fragmented, intermediate and clustered in the form of a stacked histogram (D). The percentage documented on each bar in the histogram is for the group that is seen at the maximum extent. Average mitochondrial area quantification (E) in +/+ (9 Type II NBs,3 brains), Notch[FL] (22,5), Nintra (15,4), *notch* RNAi (16,4), *su(h)* RNAi (17,4). Scale bar- 5μm. F-H: Type II NBs (mCD8-GFP, green, yellow dotted line) stained for NICD (F) and observed for distribution in +/+, Drp1[SD], *opa1* RNAi, Drp1[SD];*opa1* RNAi, *marf* RNAi, Drp1[SD];*marf* RNAi. Quantification of NICD in type II NBs (G) in +/+ (19,10), *opa1* RNAi (20,12), *marf* RNAi (15,12), Drp1[SD] (22,5), Drp1[SD];*opa1* RNAi (28,6), Drp1[SD];*marf* RNAi (11,5). Quantification of membrane NICD intensities in lineage (H, yellow arrows) in +/+ (6 lineages,3 brains), Drp1[SD] (10,4), *opa1* RNAi (9,3), *marf* RNAi (9,3), Drp1[SD];*opa1* RNAi (9,4), Drp1[SD];*marf* RNAi (8,4). Scale bar- 10μm. I: Representative confocal images of type II NB lineages showing expression of *E(spl)mγ*-GFP (green) in type II NB (large Dpn+ dotted large circles) and mature INPs (small Dpn+ dotted small circles). B, D, F & G: Graphs show mean ± sd. Statistical analysis was done by using unpaired t-test. ns- non significant, *- $p < 0.1$, **- $p < 0.01$, ***- $p < 0.001$.

suppressed in Drp1^SD;*opa1* RNAi (Fig 6F, 6G and 6H). We further assessed the activation of Notch signaling using the Notch reporter *E(spl)mγ*-GFP in the type II NB lineage. As expected, *E(spl)mγ*-GFP was found to be present in the type II NBs and mature INPs. We found that *E(spl)mγ*-GFP was present in type II NBs depleted of Opa1 (Fig 6I). However, there was a striking loss in *E(spl)mγ*-GFP expression in mature INPs (Fig 6I). The aberrant presence of Notch receptor in the cytoplasm in the type II NB did not visibly affect the activation of Notch in the NB, but Notch receptor localization and activation was abolished in the mature INPs.

Marf depletion did not show a defect in Notch receptor localization in the type II NBs and in the lineage (Fig 6F, 6G and 6H). Notch receptor distribution was also comparable to controls in Drp1^SD (Fig 6F, 6G and 6H). Marf and Drp1 depletion did not show a change in *E(spl)mγ*-GFP expression in the type II NB and lineage cells (Fig 6I).

Taken together these observations suggest that Notch accumulates in the cytoplasm and Notch pathway activation is reduced in Opa1 depleted type II NB lineages. However, unlike *notch* RNAi, there was no loss of type II NBs on depletion of mitochondrial fusion. These data show that the depletion of Opa1 does not completely inhibit Notch activity like *notch* RNAi to cause loss of type II NB numbers; rather it leads to a defect in formation and function of mature INPs thereby causing a decrease in numbers of differentiated progenies in the type II NB lineage. Loss of Notch signaling on Opa1 depletion suggests that there is a feedback between Notch signaling and mitochondrial fusion in type II NB differentiation.

## Mitochondrial fusion by overexpression of Drp1^SD induces differentiation in Notch depleted type II NB lineages

Our data shows that mitochondrial fusion on Drp1^SD overexpression did not have any differentiation defects and it also suppressed the differentiation defect in Opa1 depleted type II NB lineages. Recent studies show a role of mitochondrial fusion in proliferation of tumor NBs [24]. The transcription of Opa1 and Marf increases in *brat* RNAi and expression of Opa1 and Marf is important for proliferation of tumor NBs. We further tested whether mitochondrial fusion would drive differentiation in type II NB lineages depleted of Notch. We combined Drp1^SD to generate fused mitochondria along with *notch* RNAi, expressed this combination in the type II NB lineage and stained the brains for Dpn and Ase to count the mature INPs and GMCs and Prospero to count the young neurons (Fig 7A, 7B, 7C, 7D, 7E and 7F). The numbers of type II NBs remained less than controls in the Drp1^SD;*notch* RNAi combination and were similar to *notch* RNAi (Fig 7C). This observation further confirms that mitochondrial fusion does not affect Notch signaling in the type II NB. The loss of mature INPs (Dpn+) in *notch* RNAi expressing lineages was suppressed partially (Fig 7A and 7D). The loss of GMCs (Dpn- Ase+) in *notch* RNAi did not change in the *notch* RNAi and Drp1^SD combination (Fig 7A and 7E). The numbers of young neurons labeled with bright Pros cells also did not change significantly on expression of Drp1^SD in *notch* RNAi (Fig 7B and 7F). These results show that fused mitochondrial morphology can alleviate Notch signaling defects in type II NBs to drive the formation of mature INPs in the type II NB lineage. This suppression is partial and is not seen in the GMCs and neuron number in the lineage.

Taken together, our results show that Notch signaling regulates fused mitochondrial morphology and mitochondrial fusion is important for differentiation in the type II NB lineage. This is also consistent with the requirement of mitochondrial fusion in NB tumors induced in *brat* and *numb* mutants, since depletion of mitochondrial fusion and not mitochondrial fission abrogates the tumor phenotype in *brat* and *numb* mutant NBs [24]. In summary, we find that mitochondrial fusion controlled by Opa1 and Marf regulates Notch signaling driven differentiation in the *Drosophila* type II NB lineage.

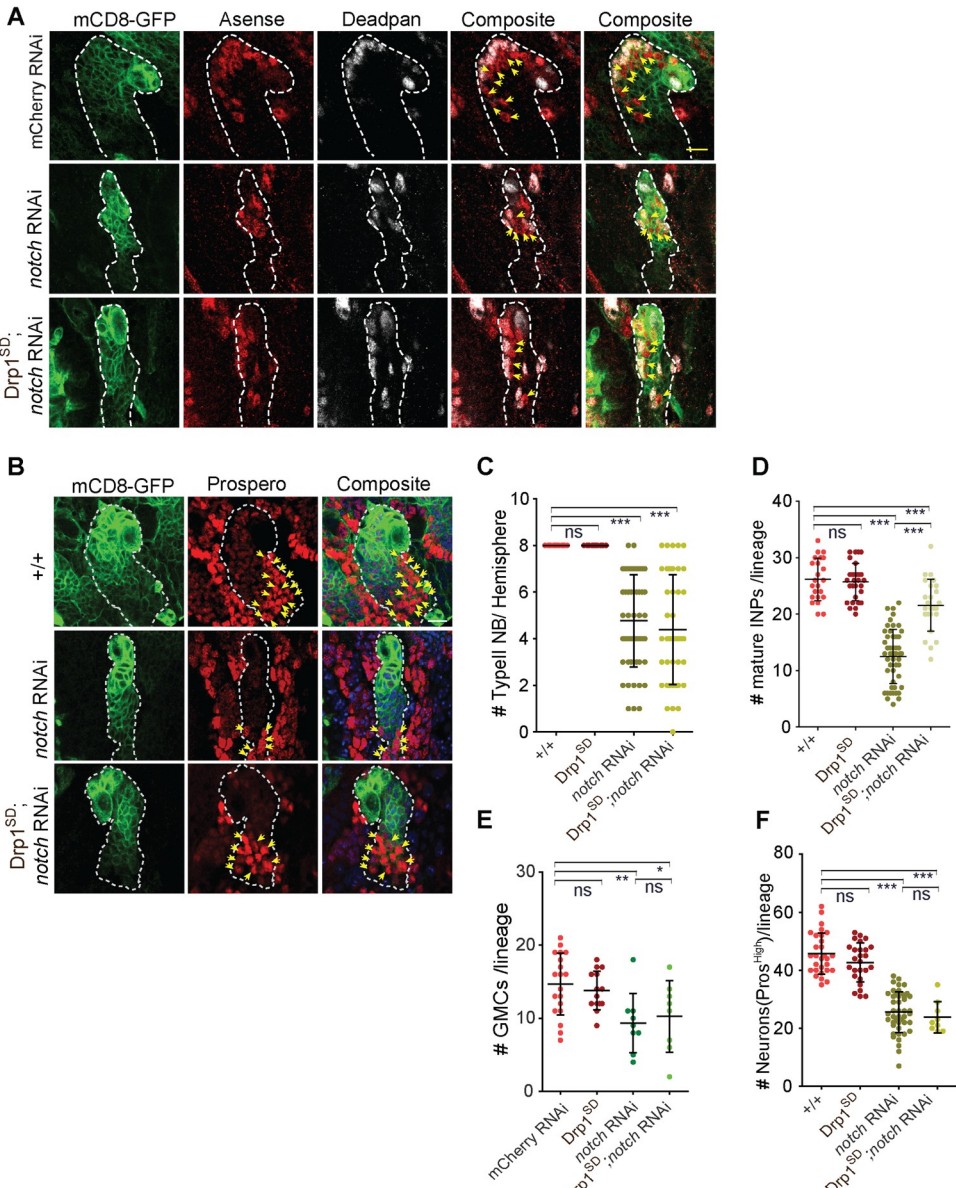

**Fig 7. Drp1^SD overexpression partially alleviates the differentiation defect seen in *notch* RNAi expressing type II NB lineages.** A, C-E: Type II lineages (*pnt*-Gal4, mCD8-GFP, green) showing Dpn positive INPs (grey) and Dpn negative Ase positive GMCs (yellow arrows) (A). Quantification of number of type II NBs (C) in +/+ (16 Brains), Drp1^SD (13), *notch* RNAi (27), Drp1^SD;*notch* RNAi (21). Quantification of Dpn positive mature INPs (D) in +/+ (23 Type II NB lineages,16 brains), Drp1^SD (28,10), *notch* RNAi (50,9), Drp1^SD;*notch* RNAi (25,5). Quantification of GMCs (E) in mCherry RNAi (19 Type II NB lineages, 6 Brains), Drp1^SD (14,5), *notch* RNAi (9,6), Drp1^SD;*notch* RNAi (8,4).Scale bar- 10μm. B,F: Type II lineages (*pnt*-Gal4, mCD8-GFP, green) showing high Pros expressing young neurons (B) (red, yellow arrows). Quantification of Dpn- Ase+ GMCs (F) in +/+ (29 NB lineages,18 brains), Drp1^SD (26,8), *notch* RNAi (45,5), Drp1^SD;*notch* RNAi (9,5). Scale bar- 10μm. C-F: Graphs show mean ± sd. Statistical analysis was done by using unpaired t-test. ns- non significant, *- p<0.1, **- p<0.01, ***- p<0.001.

## Discussion

Recent evidence shows that maintenance of mitochondrial architecture is critical for cell fate determination [65–67]. Depletion of ETC components in *Drosophila* NBs leads to mitochondrial fragmentation, loss of differentiation and cancer progression [22–24]. Here we show that

mitochondrial fusion proteins, Opa1 and Marf affect proliferation and differentiation in the type II NB lineage. Opa1 and Marf gave equivalent fragmentation and loss of mitochondrial activity in the type II NB. Opa1 and Marf depletion changes the mitochondrial density. However, decreasing Opa1 resulted in decreased proliferation of type II NBs and mature INPs and loss of mature INPs, GMCs and neurons ([Fig 8]). Decreased Marf led to an increase in the numbers of GMCs due to less proliferation thereby leading to depletion of neurons in the type II NB lineage. Interestingly, mitochondrial outer membrane fusion ameliorated the defects in mitochondrial activity and differentiation caused by Opa1 and Marf depletion. Notch signaling regulated fused mitochondrial morphology in the type II NB. Notch signaling was decreased on mitochondrial fragmentation in Opa1 depleted type II NB lineages. Mitochondrial fusion in Notch depleted type II NB lineages led to suppression of differentiation defects. Here, we discuss our results in the following contexts: 1] the mechanisms by which Notch signaling gives rise to fused mitochondrial morphology, 2] the role of the mitochondrial morphology in mediating Notch signaling and 3] the role of mitochondrial fusion in differentiation in NBs.

## Regulation of mitochondrial fusion by Notch signaling

At the heart of the discussion on interaction between Notch signaling and mitochondrial morphology, lies an analysis of how Notch signaling regulates mitochondrial fusion and activity.

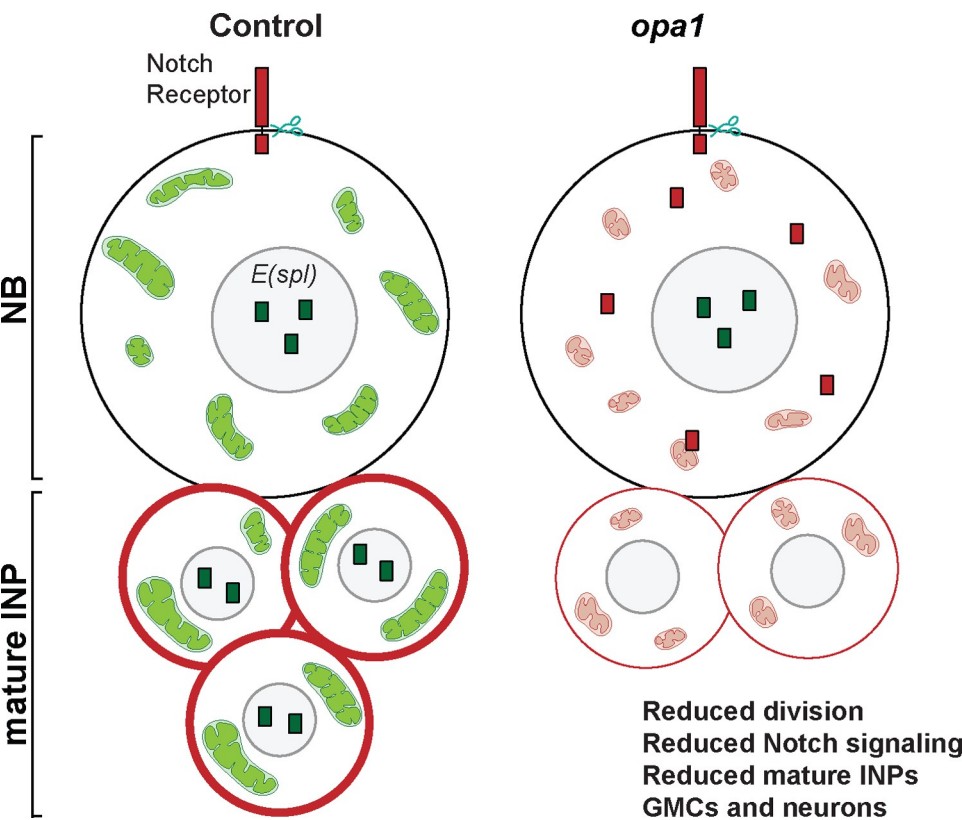

**Fig 8. Schematic summary for regulation of type II NB differentiation by Opa1 driven fused mitochondrial morphology.** Mitochondrial morphology is intermediate in state with some tubular mitochondria in the type II NBs. Notch receptor (red) is present on the plasma membrane in lineage cells and Notch signaling as seen by *E(spl)mγ*-GFP (green) is active in type II NBs and mature INPs. Loss of mitochondrial fusion by depletion of Opa1 leads to mitochondrial fragmentation in type II NBs and mature INPs, cytoplasmic accumulation of Notch receptor (red) in type II NBs and mature INPs while it is reduced from the membrane of mature INP cells leading to decrease in INP numbers.

Loss of *su(H)* also showed fragmented mitochondria similar to downregulation of Notch. In addition, mitochondrial fusion on Drp1$^{SD}$ overexpression alleviated the differentiation defects seen in type II NB lineages depleted of Notch. NICD and Su(H) are together required to activate nuclear targets downstream of Notch signaling. It is therefore possible that Notch regulates fused mitochondrial morphology through the canonical pathway by enhancing expression of fusion genes *opa1* and *marf* or regulating activity of Opa1 and Marf by post translational modification and/or reducing expression or activity of Drp1. Notch signaling may also regulate mitochondrial activity by elevating components of the ETC. Recent studies show that NB tumors arising in *brat* RNAi, which also show upregulation of Notch signaling, show increased transcription of Opa1 and Marf and mitochondrial fusion. In addition oxidative metabolism increased resulting from an increase in expression of oxidative phosphorylation enzymes to support proliferation of these tumors [24,30]. Therefore an increase in Notch activity is likely to directly affect transcription of mitochondrial morphology and oxidative phosphorylation proteins. Change in density of mitochondria by Notch signaling is also achievable by affecting the autophagy pathway [14]. It remains to be tested whether depletion of Notch leads to inhibition of transcription of mitochondrial fusion proteins or increased transcription of mitochondrial fission proteins or an impact on metabolism by decreasing autophagy.

## Impact of mitochondrial morphology on Notch signaling

Loss of Notch signaling leads to decrease in numbers of type II NBs and transition of type II NBs into type I NBs. Loss of Opa1 and Marf did not affect the numbers of type I and type II NBs. This observation suggests that mitochondrial fusion is not needed during Notch mediated formation of NBs during the embryonic stages. We found that proliferation of mature INPs is dependent upon Notch signaling driven mitochondrial fusion by Opa1. The Ase- Pros- immature INP numbers showed a variation but did not change in numbers in mitochondrial fusion depleted type II NB lineages. However, mitochondrial fusion is needed to produce healthy INPs that will mature, proliferate and differentiate to form GMCs. Further, a decrease in GMCs on Opa1 depletion occurs due to defects in division of the mature INPs. Decrease in neurons on Marf depletion arises due to decreased division rate of GMCs to neurons. Loss of mitochondrial fusion by depletion of Opa1 and Marf led to abrogation of mitochondrial activity.

How might loss of mitochondrial fusion and activity affect differentiation in the type II NB lineage? Mitochondrial morphology changes may affect levels of cell cycle regulatory molecules such as cyclin E thereby affecting Notch signaling [60,61]. This is likely to be a cause of changes in mitotic rates in the type II NB lineage. The decrease in mitotic type II NBs and mature INPs suggests a decrease in their division rate on Opa1 depletion. Marf depletion on the other hand gave an increase in dividing mature INPs. This increase in division rate of mature INPs and a decrease in division rate of GMCs causes their accumulation on Marf depletion. Further reduction in GMC proliferation rate causes depletion of Pros positive neurons on Marf depletion. Future analysis of changes in cell cycle regulatory molecules will enable us to tease apart the mechanistic reasons for differences in cell division rates on depletion of mitochondrial morphology proteins in the type II NB lineage.

Changes in mitochondrial morphology may also have an effect on the cell fate by regulating key metabolites (Choi et al. 2020; Galloway and Yoon 2013). Mitochondrial pyruvate carrier (MPC) in the inner mitochondrial membrane is involved in transport of pyruvate from cytoplasm to the mitochondrial matrix where pyruvate undergoes oxidation via tricarboxylic acid (TCA) cycle (McCommis and Finck 2015). MPC depletion in the intestinal stem cells (ISC)

results in shift of metabolism from TCA to the glycolysis which further leads to increased ISC proliferation and loss of differentiation (Schell et al. 2017). In NBs, inhibition of complex I results in loss of neuronal diversity and increased dependence on glycolysis for ATP [23,28]. Almost all glycolytic and some TCA cycle enzymes are known to localize to the nucleus in several contexts where they influence the nuclear metabolism by regulating activity of different epigenetic modifiers and cell fate [68,69]. We observed severe differentiation defects upon loss of Opa1 in NB lineage. It is possible that the defects in the inner mitochondrial membrane due to Opa1 loss affect pyruvate transport into the mitochondria which further affect NB metabolic program and differentiation. It will be interesting to check whether pyruvate metabolism is affected in the mitochondrial morphology mutants.

Our observations show that Opa1 depleted type II NBs show a loss of Notch receptors on the plasma membrane and increase in endosomes in the type II NB and in the cells of the lineage. Notch signaling is decreased in Opa1 depleted type II NB lineages. Numb regulates inhibition of Notch signaling in INPs [30]. It is possible that sustained activity of Numb in INPs and mature INPs generated from NB proliferation leads to a decrease in Notch signaling in the mature INPs. Future experiments on the defects on Numb distribution in type II NBs will reveal a mechanistic link between mitochondrial morphology, activity and Notch signaling.

Notch signaling may also be affected by interaction between mitochondrial morphology and other upstream signaling pathways. Interestingly, our finding that Notch signaling maintains fused mitochondria in NBs is in striking contrast to recent literature in other systems where fused mitochondrial morphology has been correlated with loss of *notch* activity. In triple negative breast cancer (TNBC) Notch signaling enhances mitochondrial fission via *drp1* [70]. In *Drosophila* ovarian posterior follicle cells, mitochondrial fusion induces an increase in mitochondrial membrane potential and loss of Notch signaling [18,19]. Loss of Opa1 in cardiomyocytes leads to decrease in differentiation due to enhanced Notch processivity [20]. It is interesting to speculate the reasons for observing tissue specific differences in the requirement of mitochondrial fusion for Notch signaling in differentiation. We had previously found that EGFR signaling regulates fragmented morphology for appropriate Notch signaling in *Drosophila* follicle cells [18,19]. It is likely that fragmented mitochondrial morphology leads to an interaction between EGFR and Notch signaling pathways in follicle cells and possibly other cell types such as cardiomyocytes and TNBCs where Notch signaling increases only on loss of mitochondrial membrane potential or fragmentation.

## Role of mitochondrial fusion in Notch driven differentiation in NBs

Mitochondrial fusion is coincident with elaborate cristae organization, increased activity and increased oxidative phosphorylation [47,71,72]. Drp1 mutant expression led to the formation of clustered mitochondria on one side of the nucleus, but did not show any defect on NB differentiation implying that Drp1 is dispensable for NB formation and differentiation. Interestingly, Drp1 loss driven mitochondrial hyperfusion could suppress the mitochondrial activity and differentiation defects in Opa1 and Marf depleted type II NBs. Mitochondrial fragmentation and loss of mitochondrial membrane potential is likely to cause a reduction in ATP [73]. However, ATP depletion has been reported only on combined loss of oxidative phosphorylation and glycolysis and *brat* RNAi driven tumors rely on NAD+ metabolism rather than ATP synthesis [22–24]. Since we did not see ATP stress in Opa1 and Marf depleted type II NBs, it is interesting to speculate that other mitochondrial functions dependent upon mitochondrial activity from oxidative phosphorylation are important for type II NB differentiation. Indeed mitochondrial activity may give rise to changes in calcium buffering and key metabolites [74] thereby affecting Notch signaling in NBs and lineage cells depleted of Opa1 and Marf.

Since depletion of inner mitochondrial membrane protein Opa1 affected differentiation to a greater extent as compared to Marf, it is possible that organization of the mitochondrial ETC complexes and cristae architecture in addition to fusion are crucial for type II NB proliferation and differentiation. Organization of the cristae architecture independent of oxidative phosphorylation has been previously shown to be important for *Drosophila* germ line stem cell differentiation [75]. Fusion of the outer mitochondrial membrane may restore the inner membrane organisation and increase mitochondrial activity needed in type II NB proliferation and differentiation. Recent evidence suggests that neurodegeneration defects caused by depletion of oxidative phosphorylation in Purkinje neurons can be completely rescued by mitochondrial fusion produced by overexpression of Mfn [76]. NBs deficient of the mitochondrial ETC lead to fragmentation [23] and it will be interesting to probe if fusion of mitochondria in ETC mutants will also mitigate the differentiation defect.

In summary, we find a role for mitochondrial fusion in regulation of Notch signaling in type II NB differentiation. Future studies on the mechanistic link between mitochondrial activity, change in metabolite status and Notch signaling in diverse contexts will give insight into the interaction between Notch signaling and mitochondrial morphology in a tissue specific manner. Our studies motivate an analysis of mechanisms that regulate the interaction between mitochondrial fusion or inner membrane architecture and signaling during development and differentiation at large.

## Materials and methods

### Fly genetics

Fly crosses were performed in standard cornmeal agar medium and raised 29˚C. The following fly lines were used in this study: *pntP1*-Gal4,UAS-mCD8-GFP (*pnt*-Gal4, mCD8-GFP, Jurgen Knoblich, IMP, Vienna, Austria), *elav*-Gal4, *prospero*-Gal4, *inscuteable*-Gal4, *scabrous*-Gal4, *worniu*-Gal4, *opa1* RNAi (Bloomington stock number BL32358), *opa1* RNAi (Ming Guo, UCLA), *marf* RNAi (Ming Guo, UCLA, [35]), *marf* RNAi (BL31157), *opa1* RNAi2 (BL67159), *opa1* RNAi VDRC (VDRC TID 330266), *marf* RNAi2 (BL67158, [77]), *marf* RNAi VDRC (VDRC TID 105261), *drp*1 RNAi (BL51483), *drp1* RNAi (VDRC44155), FRT40A *drp1*$^{KG}$, UASp-Drp1$^{S193D}$ (Drp1$^{SD}$ in the UASp vector, GTPase domain mutant, acts as a dominant negative on overexpression, made in the Richa Rikhy lab [31]), *notch* RNAi (BL31383), *su(H)* RNAi (BL67928), UAS-Notch$^{FL}$ (BL52309) and UAS-Nintra (LS Shashidhara, IISER, Pune, India), hSOD1 mutant (BL33607), mCherry RNAi (BL35785), UASp-mRFP (second chromosome, Richa Rikhy lab stock), mito-GFP (BL8442), *E(spl)mγ*-GFP (Rohit Joshi, CDFD, Hyderabad, India). Drp1$^{SD}$;*opa1* RNAi, Drp1$^{SD}$;*marf* RNAi, *notch* RNAi /CyOGFP; *opa1* RNAi /TM3SerGFP, *notch* RNAi/CyOGFP; *marf* RNAi/TM3SerGFP, FRT40A;*opa1* RNAi, FRT40A *drp1*$^{KG}$;*opa1* RNAi lines were generated using standard genetic crosses. *worniu*-Gal4,UAS-mCD8-GFP (*wor*-Gal4) was used to express transgenes in all the larval NBs in the third instar larval brain and *pointed*-Gal4 (*pnt*-Gal4) was used to express transgenes in the type II NBs in the larval brain. *pnt*-Gal4; mCD8-GFP crossed to Drp1$^{SD}$;mCherry RNAi, mRFP;*opa1* RNAi and mRFP;*marf* RNAi were used as controls for estimating mitochondrial morphology and differentiation of cells in the type II NB lineage in the combinations Drp1$^{SD}$;*opa1* RNAi and Drp1$^{SD}$;*marf* RNAi.

### MARCM clones for mitochondrial morphology protein mutants

MARCM clones were generated for *drp1*$^{KG}$;*opa1* RNAi and *drp1*$^{KG}$;*marf* RNAi together by the crossing *hs*-FLP; FRT 40A tub-Gal80, tub-Gal4, UAS-mCD8-GFP with FRT40A *drp1*$^{KG}$, FRT40A;*opa1* RNAi and FRT40A *drp1*$^{KG}$;*marf* RNAi respectively. First instar larvae were heat

shocked at 37˚C for 15 mins and third instar larvae obtained were checked for the presence of GFP positive clones and dissected and stained with ATPb and imaged using confocal microscopy to assess mitochondrial distribution in type II NBs.

## Immunostaining of larval brain

Wandering third instar larvae were dissected in Schneider's medium and immediately fixed in 4% PFA solution for 25 minutes at room temperature (RT). The brains were washed subsequently with 1X PBS with 0.1% Triton X-100 (PBST) for 30 minutes at RT. They were blocked with 1% BSA for 1hr at RT. The brains were stained with the appropriate primary antibody overnight at 4˚C. They were then washed 3 times with 0.1% PBST (first wash for 20 minutes and remaining for 10 minutes each). An appropriate fluorescently coupled secondary antibody was added for 1hr at RT followed by three washes with 0.1% PBST (first wash for 20 min and remaining for 10 min each) and mounted in Slow-Fade Gold (Molecular Probes).

The following dilutions were used for the primary antibodies: chicken anti-GFP (1:1000, Invitrogen), mouse-anti- ATPβ (1:200, Abcam), rabbit anti-Deadpan (1:800, Yuh-Nung Jan, UCSF, USA), rat-anti-Deadpan (1:150, Abcam), mouse-anti- Prospero (1:25, DSHB), rat-anti-Miranda (1:600, Abcam and Chris Doe, University of Oregon, USA), rat-anti-Ase (1:1000, Jurgen Knoblich, IMBA, Vienna, Austria), rabbit anti-Ase (1:10000, Yuh-Nung Jan, UCSF, USA), rabbit-anti-Cytochrome C (1:200, Cell Signaling), rabbit-anti-cleaved Caspase 3 (1:100, Cell Signaling), rabbit anti-phosphohistone 3 (1:500, Invitrogen), mouse anti-phosphohistone 3 (1:500, Invitrogen), rabbit-anti-phosphoAMPK (1:200, Cell Signaling), mouse-anti-NICD (1:10, DSHB). Hoechst (1:1000, Molecular Probes) was used to label DNA. Fluorescently coupled secondary antibodies (Molecular Probes): anti-Chicken 488, anti-Rat 568/633/647, anti-Rabbit 568, anti-Mouse 568/633 were used in 1:1000 dilution.

## DHE uptake for live imaging of ROS

Dissected third instar larval brains were treated with Dihydroethidium (DHE) (1:1000 of a 30mM stock, Molecular Probes) in Schneider's medium for 15 minutes at RT and then washed with Schneider's medium for 10 min. Brains were mounted in LabTek chambers containing Schneider's medium and imaged immediately using Zeiss LSM 710 with a 63x/1.4NA oil objective using a DPSS (561 nm) laser and dihydroethidium-1 filter settings in the Zeiss2010 software. The laser power, acquisition speed, frame size and gain were kept the same for different genotypes. The laser power and gain were adjusted to keep the range of acquisition between 0–255 on an 8-BIT scale.

## 2-Deoxy glucose treatment

Third instar larval brains were treated with 500μM 2-DG in Schneider's medium for 1hr. Control and treated brains were processed for pAMPK immunostaining as mentioned above. They were imaged at the same time with the same laser settings. The grey values increase in the 2-DG treated brains as compared to the DMSO controls.

## Mitochondrial membrane potential estimation with TMRM in live brains

Third instar larval brains were dissected in Schneider's medium. Then they were treated with Tetramethylrhodamine, methyl ester (TMRM) (100nM, Thermo fisher Scientific) for 30 mins at room temperature. Treated brains were mounted in a LabTek chamber containing Schneider's medium and imaged immediately using Zeiss LSM710 with a 63x/1.4NA oil objective. A DPSS (561 nm) laser and RFP/TRITC filter was used for the detection of TMRM signal. The

mitochondrial fusion protein depleted type II NB lineages were imaged under the same laser settings and at the same time as controls. Percentage of type II NBs showing similar TMRM intensities to the representative images were shown on the image panel for each genotype.

## Cell proliferation analysis by EdU assay

Third instar wandering larvae of selected genotypes were dissected in Schneider's medium and incubated for 1hr in 10μM EdU (5-ethynyl-2′-deoxyuridine) solution at RT followed by fixation in 4%PFA for 25min. Next brains were washed twice, 5 min each, with 3% BSA in PBS. and then permeabilized in 0.3% PBST for 20min at RT. Brains were then incubated with anti-GFP antibody in 0.3% BSA for 3hrs. After primary antibody incubation, brains were washed twice 5 min each) with PBS and then treated with 0.3% PBST for 20 min. Click-iT reaction cocktail was made according to the manufacturer's instruction. Alexa fluor 488 coupled anti-chicken antibody (1:1000, Molecular Probes) was added in reaction cocktail to detect GFP antibody added in the previous step. Brains were then treated with reaction cocktail for 30 min at RT followed by 3% BSA and 1X PBS wash for 5 min each. For DNA staining, brains were treated with Hoechst (1:1000) in 1X PBS for 10 min and then 5 min wash with 1XPBS and mounted in Slow-Fade Gold (Molecular probes) and subsequently imaged using Zeiss LSM710 with a 40x/1.4NA oil objective.

## TUNEL assay for detection of apoptotic cells

Third instar wandering larvae were dissected in Schneider's medium and fixed in 4% PFA for 25 mins followed by washing with 0.1% PBST for 20 mins at RT. Brains are then washed twice with 1XPBS for 2 min. The terminal transferase (TdT) reaction was performed as follows: First brains were treated with TdT reaction buffer and incubated for 10 min followed by treatment of freshly made TdT reaction buffer cocktail (TdT reaction buffer, 5-Ethynyl-dUTP, TdT) for 60 min at 37°C and at 500rpm. After the TdT reaction, brains were washed twice with 3% BSA at RT for 5 min. Then Click-iT reaction was performed by adding click iT reaction cocktail (Click-iT reaction buffer, Click-iT reaction buffer additive) for 30 min at RT. At this step samples were protected from light. Brains were washed with 3% BSA twice for 5 min after removing Click-iT reaction buffer cocktail and then incubated with Hoescht (1:1000) for DNA staining followed by washing with 1XPBS for 10 min. Samples mounted in Slow-Fade Gold (Molecular probes) and subsequently imaged using Zeiss LSM710 with a 40x/1.4NA oil objective and DPSS laser (561nm). Induction of apoptosis by the UAS-Hid line driven by *pnt*-Gal4 was used as positive control. TUNEL positive nuclei were counted and plotted using GraphPad Prism 5 software.

## Microscopy and Image acquisition

**Imaging of fixed samples using a confocal microscope.** Confocal microscopy of fixed samples was done at room temperature using LSM710 or LSM780 inverted microscope (Carl Zeiss, Inc. and IISER Pune microscopy facility) with a Plan apochromat 40x 1.4NA and 63x 1.4NA oil objective. Images were acquired using the Zen2010 software at 1024x1024 pixels with an averaging of 4 and acquisition speed 7. Fluorescence intensity was kept within 255 on an 8-bit scale. Following lasers were used for the excitation of different fluorophores during fixed sample imaging: Diode laser for Hoescht, Argon laser line at 488 nm for Alexa Fluor 488, DPSS laser for Alexa Fluor 568, HeNe (633 nm) for Alexa Fluor 633 and Alexa Fluor 647. The representative image for each type II lineage in the figures is shown from the center of the lineage and comprises the maximum number of cells in the lineage.

**Stimulated Emission-Depletion (STED) microscopy for imaging mitochondrial morphology.** Super-resolution microscopy was done for visualizing mitochondrial morphology within type II NBs using the Leica TCS SP8 STED 3X Nanoscope with a 100x/1.4NA oil objective. Images were acquired using the LasX software at 1024x1024 pixels to keep pixel size 20-25nm, an averaging of 4, an acquisition speed of 200 and a zoom of 4.5. The Alexa Fluor 488 and 568 were excited with Argon 488 nm and Diode 561 nm lasers respectively and emission were collected with hybrid detectors for GFP and mitochondria labeled with ATPβ antibody respectively. The 561 nm excitation laser with the 660nm depletion laser was used for stimulated emission-depletion for visualizing mitochondria by super resolution. Fluorescence intensity was kept within 255 on an 8-bit scale using the LUT mode to avoid over-saturated pixels.

## Image analysis and statistics

**Neuroblasts number analysis.** NBs in the central brain region of a single hemisphere were counted by marking with Miranda as a NB marker across different genotypes. Type II NBs were counted by using mCD8-GFP positive lineages expressed with *pnt*-Gal4. Non-parametric student's t-test was performed for statistical analysis.

**ROS intensity analysis.** Fluorescence intensity of DHE uptake in type II NB (mCD8-GFP positive) of different genotypes along with their neighboring NB (mCD8-GFP positive) were quantified by drawing a region of interest using a free hand tool in ImageJ. Ratios of DHE fluorescence intensity of GFP positive NB to a neighboring GFP negative NB present in the same optical plane were computed for all the genotypes and normalized to the average of the control to represent on a 0–1 scale. The values were plotted using GraphPad Prism software. Non-parametric student's t-test was performed for statistical analysis.

**pH3 analysis.** Total number of pH3 positive cells were counted in each type II NB lineage for control and RNAi and mutant allele expressing brains to estimate the number of mitotic cells in the lineage. Total numbers of pH3 positive NBs in the larval brain were quantified by numbers of NBs expressing pH3. Mitotic mature INPs were counted by performing Dpn staining in conjunction with pH3. Mitotic GMCs were analyzed by counting Ase, nuclear Pros and pH3 positive cells in type II lineage and plotted using GraphPad Prism software. Non-parametric student's t-test was used for statistical analysis.

**EdU analysis.** All EdU positive cells across different planes were counted in a single type II lineage for control and RNAi and mutant expressing brains and plotted using GraphPad Prism software. Non-parametric student's t-test was performed for statistical analysis.

**Quantification of mature INPs, GMCs and neurons in type II NB lineages.** The Ase and Pros negative cells adjacent to the type II NB in each lineage are immature INPs. Dpn positive cells are mature INPs. Ase staining marked immature INPs, mature INPs and GMCs. Dpn negative Ase positive cells distant from the NB and nearer to INPs are counted as GMCs in all sections in each type II NB lineage. Numbers of cells were plotted using GraphPad Prism software and non- parametric student's t-test was performed for statistical analysis.

Young neurons at the base of the lineage do not have Ase and show high nuclear Pros expression [27,78]. These bright Pros positive cells were counted from all the optical planes of the type II NB lineage to estimate the numbers of young neurons arising as a result of division of GMCs. The boundary of the lineage was identified by staining them with anti-GFP to enhance the mCD8-GFP signal. The brightness and contrast tool in ImageJ was used to further ensure that the cells being counted belong to the same lineage.

For estimating Pros positive cells in larval NBs of the type I and type II lineage together labeled with *worniu*-Gal4,UAS-mCD8-GFP, the maximum intensity optical plane was selected from anterior and posterior regions. We obtained the average area for each Pros positive

nucleus from 5 Pros positive cells in each sample. The total Pros positive area was determined by using the threshold tool in ImageJ. To extract the number of Pros positive cells, we divided the total area by the average area of a single cell positive for Pros.

## TMRM analysis

Average TMRM intensities were computed from GFP positive type II NB using a thresholding tool in the ImageJ across all the genotypes. Average intensities were then normalised to the average of the control imaged at the same laser settings. Relative intensities were then plotted using GraphPad Prism software.

## pAMPK analysis

Average pAMPK intensities were obtained from GFP positive type II NBs from different genotypes and neighboring GFP negative NBs. pAMPK fluorescence was plotted as a ratio to the neighboring control cells by using GraphPad Prism software. Larval brain treatment with 2-DG and DMSO, the brains were imaged at the same time with the same laser power and gain settings. The 2-DG treated brains showed an overall increase in antibody fluorescence and the intensity in the GFP positive type II NBs were expressed as a ratio to the average intensity of the control treated with DMSO.

## NICD fluorescence estimation

Average NICD intensities in the cytoplasm were quantified by making a rectangular ROI in the type II NB cytoplasm of different genotypes using ImageJ and then plotted as a ratio to the fluorescence in the nucleus using GraphPad Prism software. Membrane NICD intensities in the type II NB lineage were quantified by drawing ROI on the cell membrane using mCD8-GFP as a reference. Normalized NICD intensities were plotted using GraphPad Prism software.

## Mitochondrial morphology and area analysis

Brains of different genotypes were immunostained with ATPβ antibody or anti-GFP against mito-GFP to visualize mitochondria. Mitochondria appeared bead-like in control type II NBs. Mitochondria were qualitatively divided into 3 types of morphologies based on their appearance: intermediate, fused and fragmented in the type II NBs and in mature INPs [79]. A stacked histogram was plotted in Excel showing the distribution of mitochondrial morphology into these different types in Figs 1, 3, 6 and S10. The percentage phenotype that maximally occurred in each genotype is a part of images and the stacked histogram in Figs 1, 3, 6 and S10. Further, quantitative measurements for size in the form of area of mitochondria were done by using a thresholding tool in imageJ software from type II NBs. A region of interest was selected approximately in the middle of the type II NB (optical section with the highest diameter) where mitochondria are well separated. The size of the ROI was maintained the same across genotypes. Area of mitochondrial particles which were above size cut off of 0.12 $\mu m^2$ were computed and plotted using GraphPad Prism software.

## TUNEL analysis

TUNEL positive nuclei per central brain region or type II NB lineage were counted in lobes of control and *opa1* RNAi and plotted using GraphPad Prism software. We induced apoptosis by expressing UAS-Hid with *pnt*-Gal4 as a positive control for TUNEL assay.

## Supporting information

**S1 Fig. Table showing analysis of mitochondrial morphology protein knockdown with different neuronal Gal4s.** Various Gal4 drivers were crossed with RNAi depletion lines and a Drp1 dominant negative mutant of mitochondrial morphology genes at 25 or 29°C and lethality or behavioral phenotype was recorded in the adult. *worniu*-Gal4 (*wor*-Gal4), *inscuteable*-Gal4, *prospero*-Gal4 and *scabrous*-Gal4 expresses the Gal4 in all NBs, *pnt*-Gal4 expresses in type II NBs and *elav*-Gal4 expresses in neurons. Adult flies from crosses with *wor*-Gal4 (25°C) and *pnt*-Gal4 (29°C) with *opa1* RNAi and *marf* RNAi were sluggish and the numbers obtained were at the expected frequency, no lethality was seen at the pupal stage. *elav*-Gal4 crosses gave lethality and few adults emerged. The *opa1* RNAi BL32358 and *opa1* RNAi2 BL67158 gave stronger phenotypes as compared to *opa1* RNAi Ming Guo lab with *inscuteable*-Gal4, *prospero*-Gal4, *elav*-Gal4 and *scabrous*-Gal4. The *marf* RNAi Ming Guo lab and *marf* RNAi2 BL67159 gave stronger phenotypes as compared to *marf* RNAi BL31157 with *inscuteable*-gal4, *prospero*-Gal4 and *elav*-Gal4. We chose *opa1* RNAi BL32358, *opa1* RNAi2 BL67158, *marf* RNAi Ming Guo lab and *marf* RNAi BL67159 for further analysis. The Drp1 RNAi from VDRC did not show phenotypes and the Drp1 (BL51483) gave inconsistent results, hence we used a lab generated construct overexpressing the GTPase domain mutant Drp1$^{SD}$ for further analysis.
(TIF)

**S2 Fig. Depletion of mitochondrial fusion proteins Opa1 and Marf result in mitochondrial fragmentation.** A, B:Mitochondrial morphology and distribution in type II NBs (white dotted line, magnified area shown in the panel on the right) stained with ATPβ (red) antibody using STED super resolution microscopy is shown in representative images with zoomed inset in the right panel (A). Control (100% tubular, 75 NBs, 22 brains), *opa1* RNAi2 (100% fragmented, 14,4), *marf* RNAi2 (100% fragmented, 14,4). Average mitochondrial area quantification from type II NBs (B) in control (10 type II NBs, 4 brains), *opa1* RNAi2 (8,4), *marf* RNAi2 (5,3). Scale bar- 5μm. C: Clonal analysis of NB clones (green) expressing *drp1*$^{KG}$ allele show clustered mitochondria stained with ATPβ antibody (red). FRT40A *drp1*$^{KG}$ (8 NB clones), FRT40A; *opa1* RNAi (13), FRT40A *drp1*$^{KG}$;*opa1* RNAi (13). D: Representative confocal images of mitochondria labeled with mito-GFP (green) in larval muscles. Expression of *opa1* and *marf* RNAi by *mhc*-Gal4 shows smaller mitochondria while Drp1$^{SD}$ shows large mitochondria compared to control. Representative images from a minimum of 3 larvae are shown, the phenotype of smaller mitochondria is at 100% in muscle 6 and 7 of segment A2 and A3 in *opa1* and *marf* RNAi and of larger mitochondria in Drp1$^{SD}$ overexpression. Scale bar-3μm.
(TIF)

**S3 Fig. Downregulation of Opa1 and Marf does affect type II NB numbers but affect differentiated cells in lineage.** A,C:Type II NB lineages (yellow dotted line) showing expression of mCD8-GFP (green) and Dpn (red, yellow arrows) (A). Quantification of Dpn positive mature INPs (C) in control (20 NB lineages, 5 brains), *opa1* RNAi2 (19,4), *marf* RNAi2 (15,4). Scale bar- 10μm. B: Quantification of number of type II NBs in control, *opa1* RNAi, *marf* RNAi, Drp1$^{SD}$ (n = 15 brains each). C,E:Type II lineages (*pnt*-Gal4, mCD8-GFP, green) showing Pros positive INPs (red, yellow arrows) (C, yellow arrows). Quantification of Pros positive young neurons (E) in control (28 NB lineages,8 brains), *opa1* RNAi2 (19,4), *marf* RNAi2 (18,4). Scale bar- 10μm. F-G: Representative type II NB lineages showing reduced numbers of mature INPs and GMCs in *opa1* RNAi from VDRC stock center. Quantification of mature INPs (G) in mCherry RNAi, *opa1* RNAi (VDRC), *marf* RNAi (VDRC). Analysis of numbers of GMCs (Dpn- Ase+, yellow arrows) in type II NB lineage in mCherry RNAi (19 type II NB lineages, 6

Brains), *opa1* RNAi (VDRC) (20,5), *marf* RNAi (VDRC) (12,5). B,D,E& G: Graphs show mean ± sd. Statistical analysis is done using an unpaired t-test. ns- non significant, ***- p<0.001. (TIF)

**S4 Fig. NB polarity and number are unaffected by knockdown of mitochondrial morphology proteins.** A, B: Representative confocal images of larval brain lobes stained for Miranda (red) show no change in NB number (A). Expression of *opa1* RNAi, *marf* RNAi and Drp1[SD] was done by *wor*-Gal4. Quantification of NB number in larval brain lobes (B) of control (13 lobes,13 brains), *opa1* RNAi (10,10), *marf* RNAi (10,10), Drp1[SD] (10,10). Scale bar- 50µm. C: Representative images showing apical Bazooka and basal Numb localization in control and mitochondrial dynamics mutants. Bazooka is present at the apical side (away from progenies) and Numb at the basal side (near the progenies) in dividing NBs of all genotypes. Scale bar- 10µm. D, E: Representative images of brain lobes showing reduced Pros positive cells in *opa1* RNAi expressed with *wor*-Gal4, mCD8-GFP (D). Analysis of bright Pros positive cells (E) in control (10 lobes,10 brains), *opa1* RNAi (11,11), *marf* RNAi (7,7), Drp1[SD] (11,11). Scale bar- 50µm. B, E: Graphs show mean ± sd. Statistical analysis was done by using unpaired t-test. ns- non significant, ***- p<0.001. (TIF)

**S5 Fig. Expression of mRFP does not perturb the severity of *opa1* RNAi and *marf* RNAi phenotypes.** A, D: Representative confocal images of type II NB lineages showing expression of Dpn (grey scale) (A) in *opa1* RNAi, mRFP;*opa1* RNAi, *marf* RNAi, mRFP;*marf* RNAi. Quantification of Dpn positive cells (yellow arrows) (D) in *opa1* RNAi (21 Type II NB lineages, 6 Brains), mRFP;*opa1* RNAi (25,4), *marf* RNAi (15,8), mRFP;*marf* RNAi (15,3). Scale bar- 10µm. B, E: Representative confocal images of type II NB lineages showing Ase positive cells (grey scale) (B) in *opa1* RNAi, mRFP;*opa1* RNAi, *marf* RNAi, mRFP;*marf* RNAi. Quantification of Ase positive cells (yellow arrows) (E) in *opa1* RNAi (12 Type II NB lineages, 5 Brains), mRFP;*opa1* RNAi (16,3), *marf* RNAi (10,4), mRFP;*marf* RNAi (15,3). Scale bar- 10µm. C, F: Representative confocal images of type II NB lineages showing high levels of Pros expressing young neurons (grey scale) (C) in *opa1* RNAi, mRFP;*opa1* RNAi, *marf* RNAi, mRFP;*marf* RNAi. Quantification of brightly stained Pros positive young neurons (yellow arrows) (F) in *opa1* RNAi (18 Type II NB lineages, 5 Brains), mRFP;*opa1* RNAi (15,3), *marf* RNAi (15,5), mRFP;*marf* RNAi (13,3). Scale bar- 10µm. D-F: Graphs show mean ± sd. Statistical analysis was done by using unpaired t-test. ns- non significant. Values for *opa1* RNAi and *marf* RNAi are repeated from Fig 2 for comparison. (TIF)

**S6 Fig. Depletion of Opa1 does not increase pAMPK in type II NBs.** A, B: Representative images of type II NBs expressing *opa1* and *marf* RNAi along with *pnt*-Gal4, mCD8-GFP did not show change in levels of pAMPK (pAMPK fluorescence is shown as a grey scale) (A). Brains of different genotypes were treated with 2-DG and were imaged for pAMPK fluorescence at the same time under the same imaging conditions. Type II NBs are marked by white dotted lines while lineages are marked by expression of mCD8-GFP (green). Analysis of pAMPK intensity normalised with neighboring control cells (B). +/+ (32 NBs,14 brains), *opa1* RNAi (23,8), *marf* RNAi (33,8), DMSO (76,8), 2-DG (91,9). Scale bar- 10µm. (TIF)

**S7 Fig. Depletion of Opa1 shows increased ROS levels in the type II NB lineage.** A, B: Representative images showing increased levels of ROS (rainbow scale) in type II NB (position marked by white dotted line) using *pnt*-Gal4, mCD8-GFP with different genotypes (A). Analysis of relative DHE fluorescence as a ratio to neighboring cells (B) in type II NBs, +/+ (22

NBs,6 Brains), *opa1* RNAi (17,8), *marf* RNAi (17,8), Drp1$^{SD}$ (9,5), hSOD1 mutant overexpression (13,3). Scale bar- 10μm.
(TIF)

**S8 Fig. Depletion of Opa1 and Marf does not cause increase in the cleaved Caspase 3 levels and apoptosis in the brain.** A: Representative images of type II NB lineages (green) stained for cleaved caspase 3 and shown in heatmap. +/+ (for *opa1* RNAi, 10 NB lineages, 6 brains), *opa1* RNAi (9,6), Control (for *marf* RNAi, 24 Type II NB lineages, 6 Brains), *marf* RNAi (30,5). Scale bar- 10μm. B, C: Fluorescence confocal images of brain hemispheres showing no significant change in TUNEL positive nuclei (red, white arrows) (B) in *opa1* RNAi expressed in all NBs with *wor*-Gal4. Scale bar- 50μm. Expression of hid shows a significant increase in TUNEL positive cells when expressed with *pnt*-Gal4, mCD8-GFP in the type II NB lineage. Quantification of TUNEL positive nuclei (C) in *wor*-Gal4/+ (3 brains), *wor*-Gal4 *opa1* RNAi (4), *pnt*-Gal4 UAS-*hid reaper* (5). Statistical analysis was performed using unpaired t-test. ns- non significant, ***- p<0.0001.
(TIF)

**S9 Fig. Analysis of proliferation in NBs depleted of Opa1 and Marf.** A-B: Representative confocal images of larval brain lobes showing reduction of pH3 positive NBs upon loss of Opa1 (A). Analysis of numbers of pH3 positive NBs (yellow arrows) in larval brain lobe in control (8 brain lobes), *opa1* RNAi (9), *marf* RNAi (6). Scale bar- 50μm. C: Quantification of total number of pH3 positive cells, and percentage of dividing mature INPs in type II NB lineages in mCherry RNAi (33 Type II NB lineages,7 brains), Drp1$^{SD}$;mCherry RNAi (24,6), *opa1* (33,6, Drp1$^{SD}$;*opa1* RNAi (26,8), *marf* RNAi (33,8), Drp1$^{SD}$;*marf* RNAi (15,5). D: Representative images of type II NB lineages labeled with CD8-GFP (green), Pros (blue), Ase (red), pH3 (grey) showing reduced mitotic GMCs (yellow arrows, pH3+light Pros+Ase+) in *marf* RNAi (in support of graph presented in Fig 5D). Scale bar- 10μm.
(TIF)

**S10 Fig. Notch signaling regulates fused mitochondrial morphology in INPs and muscle cells.** A: Representative confocal images of type II NB lineage (yellow dotted line) showing expression of mCD8-GFP (green) and Dpn (red) in the NBs. Note that Dpn is present in the NB and in the mature INPs in the lineage +/+ (39 Type II NBs, 5 brains), Notch$^{FL}$ (>100, 5). B: Representative confocal images of type II NB lineage (yellow dotted line) showing expression of mCD8-GFP (green) and Ase (red) in the type II NB lineage. Note that Ase is not present in the NB but present in the lineage +/+ (31 Type II NBs, 5 brains), Notch$^{FL}$ (>100, 5). C-D: Larval brain lobes show suppression of Notch mediated NB hyper proliferation on depletion of Opa1 and Marf (C). Quantification of NB number (D) in *pnt*-Gal4, UAS-mCD8-GFP control (14 lobes), Notch$^{FL}$ (11), Notch$^{FL}$;*opa1* RNAi (9), Notch$^{FL}$;*marf* RNAi (9). Scale bar- 50μm. E-F: Confocal images showing mitochondria (red, stained with ATPβ) in mature INPs of type II NBs expressing mCherry RNAi (40 mature INPs, 5 brains, 8 lineages, 75% intermediate), *notch* RNAi (49 mature INPs, 4 brains, 10 lineages, 69% fragmented), *su(H)* RNAi (11 mature INPs, 3 brains, 4 lineages, 81% fragmented) and N$^{FL}$ (68 mature INPs, 4 brains, 13 lineages, 69% clustered). The control cohort is the same as that used in Fig 1. Graph shows the distribution of mitochondria into fragmented, intermediate and clustered in the form of a stacked histogram (F). The percentage documented on each bar in the histogram is for the group that is seen at the maximum extent. Scale bar- 2.7μm. G: Representative confocal images of mitochondria labeled with mito-GFP in muscle cells showing smaller mitochondria in *notch* and *su(H)* RNAi and larger mitochondria on Nintra overexpression. The image for control mCherry RNAi is repeated from S2 Fig for comparison. Representative images from a

minimum of 3 larvae are shown, the phenotype of smaller mitochondria is at 100% in muscle 6 and 7 of segment A2 and A3 in *notch* and *su(H)* RNAi and of larger mitochondria in Nintra overexpression. Scale bar-3μm.
(TIF)

**S1 Data. File containing data underlying the plots in the Fig 1.**
(XLSX)

**S2 Data. File containing data underlying the plots in the Fig 2.**
(XLSX)

**S3 Data. File containing data underlying the plots in the Fig 3.**
(XLSX)

**S4 Data. File containing data underlying the plots in the Fig 4.**
(XLSX)

**S5 Data. File containing data underlying the plots in the Fig 5.**
(XLSX)

**S6 Data. File containing data underlying the plots in the Fig 6.**
(XLSX)

**S7 Data. File containing data underlying the plots in the Fig 7.**
(XLSX)

**S8 Data. File containing data underlying the plots in the S2 Fig.**
(XLSX)

**S9 Data. File containing data underlying the plots in the S3 Fig.**
(XLSX)

**S10 Data. File containing data underlying the plots in the S4 Fig.**
(XLSX)

**S11 Data. File containing data underlying the plots in the S5 Fig.**
(XLSX)

**S12 Data. File containing data underlying the plots in the S6 Fig.**
(XLSX)

**S13 Data. File containing data underlying the plots in the S7 Fig.**
(XLSX)

**S14 Data. File containing data underlying the plots in the S8 Fig.**
(XLSX)

**S15 Data. File containing data underlying the plots in the S9 Fig.**
(XLSX)

**S16 Data. File containing data underlying the plots in the S10 Fig.**
(XLSX)

## Acknowledgments

We thank the RR lab members for discussions on this project and feedback on the manuscript. We thank the *Drosophila* facility and microscopy facility at IISER, Pune, India for support

throughout this project. We thank Juergen Knoblich (IMP, Vienna) for the rat anti-Asense antibody and Chris Doe for the rat anti-Miranda antibody (University of Oregon, USA).

## Author Contributions

**Conceptualization:** Dnyanesh Dubal, Richa Rikhy.

**Formal analysis:** Dnyanesh Dubal, Prachiti Moghe, Rahul Kumar Verma, Bhavin Uttekar, Richa Rikhy.

**Funding acquisition:** Richa Rikhy.

**Investigation:** Dnyanesh Dubal, Prachiti Moghe, Rahul Kumar Verma, Bhavin Uttekar, Richa Rikhy.

**Methodology:** Dnyanesh Dubal, Prachiti Moghe, Rahul Kumar Verma, Bhavin Uttekar, Richa Rikhy.

**Project administration:** Richa Rikhy.

**Supervision:** Richa Rikhy.

**Validation:** Dnyanesh Dubal, Prachiti Moghe, Rahul Kumar Verma, Bhavin Uttekar, Richa Rikhy.

**Visualization:** Dnyanesh Dubal, Prachiti Moghe, Rahul Kumar Verma, Bhavin Uttekar, Richa Rikhy.

**Writing – original draft:** Dnyanesh Dubal, Richa Rikhy.

**Writing – review & editing:** Dnyanesh Dubal, Prachiti Moghe, Rahul Kumar Verma, Bhavin Uttekar, Richa Rikhy.

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
