## [Decision Letter · Decision Letter 0]

20 Jan 2021

Dear Dr Rikhy,

Thank you very much for submitting your Research Article entitled 'Mitochondrial fusion regulates proliferation and differentiation in the type II neuroblast lineage in Drosophila' to PLOS Genetics.

The manuscript was fully evaluated at the editorial level and by independent peer reviewers. The reviewers appreciated the attention to an important problem, but raised some substantial concerns about the current manuscript. Based on the reviews, we will not be able to accept this version of the manuscript, but we would be willing to review a much-revised version. We cannot, of course, promise publication at that time.

If you decide to revise the manuscript for further consideration at PLOS Genetics, please aim to resubmit within the next 60 days, unless it will take extra time to address the concerns of the reviewers, in which case we would appreciate an expected resubmission date by email to plosgenetics@plos.org.

[LINK]

We are sorry that we cannot be more positive about your manuscript at this stage. Please do not hesitate to contact us if you have any concerns or questions.

Yours sincerely,

Hongyan Wang, Ph.D.

Associate Editor

PLOS Genetics

Gregory P. Copenhaver

Editor-in-Chief

PLOS Genetics

Reviewer's Responses to Questions

**Comments to the Authors:**

Reviewer #1: In this study, Dubal et al. studied potential roles of mitochondria dynamics in normal development of Drosophila type II neuroblast lineages. Specifically, they examined how knocking down mitochondria fusion/fission proteins Opa1, Marf, and Drp1 would affect the morphology of mitochondria in type II neuroblasts, generation of INPs and GMCs. Further, they examined functional relationship between Notch signaling and mitochondria dynamics in type II neuroblast lineage development. Previous studies have shown that oxidative phosphorylation is required for the growth/proliferation of Drosophila tumorigenic neuroblasts, but it seems still a little controversial whether oxidative phosphorylation is required for normal development of type II NB lineages. Therefore, it is interesting to directly examine whether manipulating mitochondrial fission and fusion, which are related with oxidative phosphorylation, would affect normal development of type II NB lineages. However, although this manuscript addresses an interesting question, I have quite a few major concerns that significantly dampen my enthusiasm for this work.

1. A recently published paper (Bonnay F, et al, Cell, 2020, 182: 1490) also examined how knockdown of Marf or Opa1 would affect normal type II neuroblast development and their results showed that knockdown of Marf or Opa1 resulted in a similar or increased number of INPs, which is contradictory to the results described in this manuscript. The authors should address the discrepancy.

2. Regarding the mitochondria morphology, I am really not sure if anyone could easily see the difference in different groups in Fig 1B, Fig 3A, Fig6C, fig S2A, and FigS5D without looking at the quantification data except that it is clear that expressing Drp1SD leads to aggregation of mitochondria on one side of the NB. They all appear spherical to me. At least I don’t see the clear difference between spherical and elongated/tubular mitochondria structures as shown in some published papers. In fact, mitochondria should mostly appear spherical in normal type I and type II NBs according to published papers rather than tubular as described in this manuscript. The authors may want to confirm their results using different mitochondria markers to verify their findings. Furthermore, the authors should measure the length of the mitochondria for quantifying the mitochondria phenotypes. Also, I am not sure if anyone could count the number of mitochondria accurately with the resolution and quality of the images shown in the manuscript. Personally, I don’t think so and I don’t have confidence in the quantification data of the mitochondrion numbers shown in this manuscript. I don’t think it is extremely important to quantify the number of mitochondria (unless they can truly distinguish individual mitochondria, but that obviously is not the case as they stated that they only quantified the area of “optically resolvable” mitochondria, meaning that at least some mitochondria are not optically resolvable in their images). What matters is the morphology (the area and length) of the mitochondria. If the authors are not totally confident about the quantification data of the mitochondria numbers, these bar graphs should be simply removed.

3. The authors counted wrong cells for quantifying the number of GMCS for Fig 2E, Fig 3G, Fig 7E, Fig S2G, Fig S3F. The authors used Pros staining as a single marker for identifying GMCs. However, Pros is not only expressed in GMCs but also neurons (particularly young neurons). The expression of nuclear Pros in GMCs is usually weaker than that in neurons. The authors counted mostly the cells with strong Pros staining as GMCs, but actually these cells mostly are neurons rather than GMCs. To quantify GMCs, they need to use additional makers such Elav or Ase. GMCs express nuclear Pros and Ase but not Elav or Dpn. In addition, I don’t think their quantification of the number of immature INPs in fig 2C is totally accurate. The number of immature INPs never exceed 10 per lineage in normal brains, but they have several samples that have more than 10 imINPs per lineages. The authors have to make sure not to include cells with weak nuclear Pros staining as imINPs or they could other markers such PntP1 staining to label imINPs. Further, even if the authors use the Pros staining for quantifying the number of neurons, the quantification results may not be reliable either because mCD8-GFP driven by pntP1-GAL4 can not label all cells in type II NB lineages.

4. The authors showed that Marf RNAi knockdown led to a reduction in the number of Pros+ cells in type II NB lineages but did not affect the number of imINPs, mINPs, or pH3+ cells in the lineages. Logically it does not make sense unless that there is an increase in apoptotic cell death of postmitotic neurons, which seems not likely based on what they found with Opa1 knockdown. The authors may want to take a look at apoptosis after knocking down Marf and discuss why there is a reduction in Pros+ cells after knocking down Marf.

5. In Fig 6, the data show that knockdown of Opa1 leads to a reduction in the number of pH3+ cells in type II NB lineages but expressing Drp1SD does not, however, when Opa1 RNAi and Drp1SD are expressed simultaneously, there is an increase in the number of pH3+ cells. Logically it does not make sense. Same concern for the increased number of pH3+ cells resulting from the simultaneous expression of Marf RNAi and Drp1SD. The authors may want to double check their quantification data to make sure they are accurate.

6. I do not find data shown in Fig 6 are particularly informative. The purpose of this figure is to investigate why knocking down of Opa1 or Marf leads to reduction in the number of mINPs and/or Pros+ cells. They need to look at the mitotic rate of individual cells types including type II NBs, mINPs, and GMCs separately rather than just count the total number of pH3+ cells in the lineages, which does not tell whether there is any defect in proliferation of type II NBs, mINPs or GMCs. With the data they have right now, they cannot draw the conclusion that “opa1 depletion decreased the rate of mINP proliferation” (at the end of 1st paragraph on page 11).

7. There is inconsistency between what the authors described and the literature they cited. For example, at the beginning of the second paragraph on page 5, they described that mitochondria are tubular and cited the reference #31. However, in that reference, it is clearly stated that mitochondria in NBs appear spherical, which is also confirmed in other publications such as the reference #24 they cited.

8. In Fig S3 (the 3rd paragraph on page 6), the authors used woriu-GAL4 to knockdown Opa1, Marf, or Drp1 and labeled the brains with Miranda staining for quantifying the number of type I and type II NBs. Miranda is expressed in both type I and type II NBs. Without any additional markers, how could the authors distinguish type I vs type II NBs?

9. For Fig S4A-B, I am not convinced that there is any positive pAMPK staining signal in the NBs. They all appear similar to the background.

10. In Fig 6A-B, the authors quantified the number of mINPs by counting the number of Dpn+ cells. However, expressing NotchFL would lead to dedifferentiation of imINPs and generation of extra type II NBs. They need to include additional Ase staining to ensure they are not counting the dedifferentiated type II NBs after expressing NotchFL.

11. In Fig 6C-E, the authors found that expressing NotchFL led to an increase in the mitochondria area and a reduction in the number of mitochondria. However, when Nintra was express, they found that the area of mitochondria was not affected but the number was reduced. Nintro is supposed to have stronger effect than NotchFL. It has been shown in the reference #24 that knocking down Numb, which will lead to over-activation of Notch signaling like the expression of Nintra, led to fusion of mitochondria and that this phenotype is a common feature of tumorigenic NBs as knockdown of Brat also led to a similar fusion of mitochondria. Therefore, the results are contradictory to published data. It seems unlikely that expressing Nintro would not lead to tubular mitochondrial morphology and an increase in the area of mitochondria.

12. The loss of mINPs in Notch RNAi type II NB lineages is at least in part to due to ectopic Ase expression in type II NBs and subsequent transformation of type II NBs into type I NB. If Drp1SD expression partially rescued the number of mINPs and Pros+ cells in Notch RNAi type II NB lineages as shown in Fig 7, does the Drp1SD expression also suppress the ectopic Ase expression in Notch RNAi type II NBs? If mitochondrial fusion induced by the expression of Drp1SD could not rescue the loss of type II NBs resulting from Notch RNAi, why could it alleviate notch signaling defects in mINPs and rescue the loss of mINPs? Could the author provide any reasonable explanations? Also, in Fig 7D, it seems that Pros staining did not work well for Drp1SD Notch RNAi group. The staining pattern is very different from those in other panels. Dpn staining for NotchFL Marf RNAi in Fig7F did not go well either in Fig7F. It looks like just background. The author should redo the staining of Pros and Dpn for these two groups. Further, co-staining of Ase should be done for Fig7F to unambiguously distinguish type II NBs from mINPs because the expression of NotchFL causes dramatic increase in the number of type II NBs due to dedifferentiation of imINPs and some dedifferentiated NB may still remain small and are difficulty to be distinguished from mINPs based on the cell size.

13. The authors need to describe clearly what they mean by “differentiation of type II NB lineages” when they use this term to describe the phenotypes. This is a very vague term and could mean a lot of things, such as differentiation of NBs, differentiation of INPs, neuronal identity specification, etc. They need to be more specific when describing the phenotypes. They need to make it clear the phenotypes are defects in type II NB proliferation/self-renewal, INP maturation/proliferation/self-renewal, GMC production/division, or neuronal production/fate specification. The authors also should not use the term “mutant” for RNAi knockdown. They are different.

14. The subtitle on page 12 “Mitochondrial fusion induces differentiation on Notch depletion in the type II NB lineage” is very confusing. Did the author mean “Mitochondrial fusion induces differentiation of Notch-depleted type II NB lineages”?

Reviewer #2: Dubal et al., shows in this ms that mitochondria architecture is important for differentiation of type II lineages. They show that Drp1 (fission protein) has a different phenotype to the of Opa1 and Marf (inner membrane fusion protein and out membrane protein) respectively. The latter two shows loss of mitochondria activity and failure for type II NBs to differentiate (just like notch mutant). OPa1 is also required for MMP and caused increased cytochrome c in type II NBs when inhibited. Opa1 depletion decreased the rate of mINP proliferation in the type II NB via decreasing its rate of cell cycle progression. Finally, the authors show that inhibiting Notch RNAi causes failure to differentiate, and this can be rescued by Drp1 dominant negative. The work is of general interests to the field of developmental neurobiology, as it shows that fission and fusion of the mitochondria is important for NB differentiation.

Major points:

I see two main issues with this ms. First is the quantification of mitochondria fusion/fission phenotype, the images are not very obvious to interpret, and how quantification can reflect these phenotypes is not so clear. Either representative 3D reconstruction should be provided, or clearer explanation of how quantification is done should be provided.

The second issue is regarding the discrepancy of Notch activated vs Notch RNAi phenotype, and why Notch intra/Full length is rescued by Opa1 and Marf in terms of NB number, and conversely why Drp1 DN rescues the differentiation phenotype (which makes a lot more sense than the former).

1. “Opa1, Marf and Drp1 depletion by multiple RNAi lines and mutants showed survival of animals until the pupal stage with inscuteableGal4 and worniu-Gal4 and were lethal or showed behavioral defects as adults.”MarfRNAi seems to be published else where, is Opa1 and Drp1 RNAis validated? Should be done with qPCR.

2. It is not clear to me how mitochondria number is estimated in Fig 1B and S2, as the mitochondria punctas are not well separated. In the methods, it is stated that watershed tool is used, however, does this work, if all objects are touching? Additional more convincing images are needed to support this data.

3. What are the weakly positive pros cells in Figure 2 S2 and Figure 3? How do you make conclusions that it is GMCs and not INPs, if Dpn is not used in conjunction in the same sample?

4. Notch RNAi causes mitochondria fragmentation and by suppressing Drp1, then differentiation (not nb number) is rescued significantly. To better quantify the differentiation phenotype, overall clone volume of type II lineages under the genetic manipulations should be estimated.

5. TMRM intensities were computed in the entire type II NB in control, opa1 RNAi and marf RNAi expressing brains and in the clustered mitochondria in Drp1SD, Drp1SD;opa1 RNAi and Drp1SD;marf RNAi expressing brains using ImageJ. Why different parameters?

6. Notch intra full length gives ectopic NBs, and Opa1 and Marf1 RNAi surpresses the fusion, and somehow affects NB number. How this happens is not clear to me. The authors suggests it is via affecting NB proliferation, but it could also be due to affecting the rate of INP to NB dedifferentiation, the authors should show which is the case.

Minor points:

S3 was cited before S2 D,E, citing of S figures are all over the place.

“Our data together shows that mitochondrial morphology is tubular in type II NBs and fusion of mitochondria alone did not give a differentiation defect but was able to suppress the differentiation defect in opa1 mutant NBs”. This is a long sentence, perhaps needs rewriting.

Reviewer #3: In this manuscript, Dubal et al investigated the role of mitochondrial morphology during differentiation of type II neuroblasts in Drosophila. The authors performed genetic and imaging experiments to assess the effect of blocking mitochondrial fusion or fission on neuroblast proliferation, differentiation and viability. They found that depletion of Opa1 results in changes in mitochondrial network and reduced neuroblast differentiation. The authors further showed the differentiation defect induced by knocking down Opa1 is not due to cell death, but more likely the consequence of a reduced proliferation rate of neuroblast. Finally, the authors investigated how Notch signaling intersects with mitochondrial morphology in regulating neuroblast differentiation. Although many findings including those showing the effect of knocking down fusion & fission proteins on the morphology of mitochondria are not novel as they have been shown extensively by others, some data could be a worthwhile contribution to the field of stem cell mitochondrial dynamics. Nevertheless, there are a number of conclusions that require additional support and controls to be rectified.

Major comments

1. The authors claimed that the phenotypic effects of Opa1-RNAi can be rescued by combining opa1 depletion with the overexpression of a Drp1 dominant negative construct (Drp1sd), which is presumably UAS-Gal4 based (the authors need to specify the nature of the construct in the manuscript somewhere, see below). Theoretically, Drp1sd should promote fusion to counteract Opa1 RNAi induced fission. They also use a similar approach for other RNAi lines (e.g. Marf1 RNAi and notch RNAi) and also combine NotchFL with Opa1 or marf1 RNAi. However, the authors compare these combined genotypes to Drp1SD (or NotchFL) alone, when the correct control would be in combination with another RNAi that has no phenotypic effect (e.g. mCherry RNAi). This is because an increased number of UAS-driven constructs within a genotype can reduce their overall expression by limiting available Gal4. Many of the effects they see when comparing combined genotypes to single genotypes could be explained by such a dosage effect. Therefore, a correct control must be included to draw conclusion based on these experiments. What’s more, because Drp1sd is a novel line, the manuscript would benefit from further details about this line in the methods section, such as the vector used to clone the Drp1sd (e.g. UASt or UASp), into which chromosome this construct was inserted? I also wonder why Drp1sd overexpression is used in combination with Opa1/Marf RNAi instead of combining DRP1 RNAi and Opa1/Marf RNAi to achieve double knockdown. Based on Table S1, Drp1 RNAi seems to work well as it gave a similar defect at the organismal level.

2. Despite that the authors used the STED super-resolution imaging to visualize the mitochondrial network in neuroblasts and differentiated cells, the quality of the images is not impressive. It is hard to tell whether those mitochondria in Drp1sd overexpressing cells or notch down-regulating are fused, or simply it is an aggregation of many small mitochondria (e.g. Fig 1B, Figure 3A & Figure 6 etc). A number of images were over-exposed with over-saturated signals, and this will affect the qualifications presented in those figures.

3. Given that Opa1 and Marf knockdown gave a different level of differentiation deficiency, and Opa1 has the additional role in cristae remodelling (which is directly linked to electron chain transport capacity) besides being a mitochondrial fusogen like Marf, it will be great to show the cristae structure by electron microscope after Marf and Opa1 knockdown. However, I understand that performing such additional experiments could be challenging during the pandemic.

4. Page 13, 2nd paragraph, the authors state: In summary our results show that mitochondrial fusion controlled by Opa1 and Marf regulate Notch signaling driven differentiation in the Drosophila. I feel the rescue data in Fig 7 are not sufficient to support such a statement. Does Opa1 and Marf knockdown reduce notch signaling? Does notch knockdown lead to decreased Opa1 and Marf expression? In the discussion, the authors stated: our preliminary analysis shows that Opa1 mutant type II NBs show a loss of notch receptors on the plasma membrane and increase in endosomes in the type II NB and the lineage (data not shown). It is an important piece of data to support the link between notch and mitochondrial fusion proteins. Why these data are not shown to further support the statement made in the last paragraph of the discussion, which is “we find a distinct role for mitochondrial fusion in regulation of notch signaling in type II NB differentiation”.

5. At multiple points in the manuscript, the authors used the term “mutants” to refer to RNAi lines or Drp1sd overexpression, which is not correct. Please use a more accurate phrase such as “Opa1 or Marf depleted genotypes” or “combined genotypes”, as relevant.

Minor comments

1. Some of the panels in supplementary figures could be moved into different figures to aid the flow. Specifically, S2D-G would be better suited in S3 and S4E-G would be better suited as a separate figure.

2. Page 5, 1st paragraph, “multiple RNAi lines and mutants” should be replaced by “multiple RNAi lines and overexpression of a dominant-negative Drp1 mutation”.

3. Page 7, 3rd paragraph, which Gal4 did the authors use to drive the expression of RNAi and Drp1sd overexpression? Please specify.

4. Page 11, 1st paragraph, “also showed significant increase in PH3” should be replaced by “also showed a significant increase in PH3”.

5. In the first paragraph of the section “Mitochondrial fusion induces differentiation on Notch depletion in the type II NB lineage”, please briefly describe relevance of Nintra whilst introducing Notch signaling.

6. Page 11, 2nd paragraph, do the authors intended to say “Notch signaling is activated in type II NBs and suppressed in immature INPs.”?

7. Figure 1B – These images would benefit from also showing nuclear staining if the authors already have already done so, because the high background of CD8-GFP makes delineation of the neuroblast difficult to distinguish.

8. Figure 1C - A limitation of this quantification of mitochondrial area and number is that these measurements were done on a single plane. As it is difficult to overcome this limitation, it would be useful to have this clarified in the figure legend.

9. The final paragraph of the section “Mitochondrial fusion is required for differentiation in the type II NB lineage” would be more suitable for the discussion, so should be moved or shortened.

10. Figure 3D – The same control images has been used for Figure 2 and 3. It therefore may be more appropriate to group Figures 2 and 3 rather than repeat data.

11. Figure 4A – What the numbers on the panels represent has not been stated or, if stated, should be made clear.

12. Figure 4B and associated text - Please expand on how neighboring cells were selected and identified for comparison (also for S4B and S4D).

13. Figure 7G – There is inconsistency to other figures because the NB number has not been normalized per hemisphere. Is there a reason for this?

**Have all data underlying the figures and results presented in the manuscript been provided?**

Reviewer #1: **No: **

Reviewer #2: Yes

Reviewer #3: Yes

PLOS authors have the option to publish the peer review history of their article (what does this mean?). If published, this will include your full peer review and any attached files.

Reviewer #1: No

Reviewer #2: No

Reviewer #3: No

---

## [Decision Letter · Decision Letter 1]

20 Jul 2021

Dear Dr Rikhy,

Thank you very much for submitting your Research Article entitled 'Mitochondrial fusion regulates proliferation and differentiation in the type II neuroblast lineage in Drosophila' to PLOS Genetics.

The manuscript was fully evaluated at the editorial level and by independent peer reviewers. While reviewers agreed that the revision was improved, they shared similar concern that the quantification of the individual mitochondrion and labeling of distinct cell types within neuroblast lineages were not convincing. Based on the reviews, we will not be able to accept this version of the manuscript, but we would be willing to review a much-revised version. We cannot, of course, promise publication at that time.

If you decide to revise the manuscript for further consideration at PLOS Genetics, please aim to resubmit within the next 60 days, unless it will take extra time to address the concerns of the reviewers, in which case we would appreciate an expected resubmission date by email to plosgenetics@plos.org.

[LINK]

We are sorry that we cannot be more positive about your manuscript at this stage. Please do not hesitate to contact us if you have any concerns or questions.

Yours sincerely,

Hongyan Wang, Ph.D.

Associate Editor

PLOS Genetics

Gregory P. Copenhaver

Editor-in-Chief

PLOS Genetics

Reviewer's Responses to Questions

**Comments to the Authors:**

Reviewer #1: The manuscript has improved to some extent and authors have address some of my concerns. However, some critical concerns I had before still have not been addressed. Some of their conclusions are still not well supported by their data and some data could be even mis-leading. Given that there are still so many concerns/problems after revision, I would not support publication of the manuscript in PLoS Genetics.

Here are some of the major concerns that still remain to be addressed (most of these concerns are the ones I mentioned previously).

1. The authors showed that Opa1 knockdown led to a reduction in the number of Dpn+ cells, Ase+ cells, and Pros [high] cells. By co-staining of pH3, they further showed that mitotic rate of mINPs (Dpn+ cells) was reduced. Based on these data, the author concluded that the reduction of Dpn+ cells was because of the reduced mitotic rate of mINPs. However, if mINPs have a reduced mitotic rate, there should be an increase in the number of mINPs (as they argued that reduction in the mitotic rate likely accounts for the increased number of GMCs in Marf knockdown lineages) unless the number of cell cycles that mINPs will undergo is reduced and mINPs terminate their self-renewal prematurely. They have to look at the mitotic rate of type II NBs upon knockdown of Opa1 (and Marf knockdown as well) as I suggested previously. Reduction in the mitotic rate of type II NBs certainly will also lead to a reduction in the number of mINPs and more likely is the real reason.

2. The authors showed that Marf knockdown resulted in an increased mitotic rate of Dpn+ cells (mINPs), an increased number of Ase+ cells, and a reduction in the number of Pros[high] cells, without changes in the number of mINPs. They concluded that the number of GMCs was increased but GMCs failed to produce neurons or produced much fewer neurons. They offered two different explanations for the increase in the number of GMCs and reduction in the number of Pros[high] cells: increased proliferation of mINPs and reduced proliferation of GMCs in the results and discussion parts. There are many problems here. First, if mINPs divide more quickly than usual, then the number of Dpn+ mINPs should be reduced unless mINPs can undergo more cell cycles than wild type mINPs. They should provide data to show if the number of cells cycles of mINPs is changed or not (e.g. by examining the number of neurons generated in an INP clone). Second, if they think that the number of GMCs was increased after Marf knockdown, they need to do Ase and Pros co-staining rather than Ase staining alone to un-ambiguously identify GMCs for quantifying the number of GMCs. Third, they need to examine the mitotic rate of GMCs specifically by co-staining Ase, Pros, and pH3, if they suspect that GMCs do not divide as actively as usual after knockdown of Marf. It seems odd that knockdown of Marf would increase the proliferation of mINPs but decrease the proliferation of GMCs. How could knockdown of Marf have totally different effects in INPs and GMCs in terms of cell proliferation?

3. Quantifying the number of neurons in type II NB lineages by counting the number of Pros[high] cells in type II NB lineages labelled with mCD8-GFP driven by pnt-GAL4 is not a proper way for quantifying the number of neurons and the results could be misleading. The reason is because pnt-Gal4 is only active in type II NBs and imINPs and mCD8-GFP expression in the progeny decreases gradually and dose not label all the cells produced in individual type II NB lineages. The expression of mCD8-GFP driven by pnt-GAL4 does not tell where is the boundary of the lineage. They have to label the lineages with mCD8-GFP driven by either elav-GAL4 or tub-GAL4 and compare the number of neurons in MARCM clones or flipout clones.

4. Throughout the entire paper, the mitochondria morphology is examined in type II NBs, however, the proliferation/differentiation defects were mostly observed in INPs/GMCS/neurons. Are the defects in mitochondrial morphology observed in type II NBs resulting from Opa1/Marf/Drp1 knockdown/inhibition or Notch manipulations really responsible for the proliferation/differentiation defects of INPs/GMCs? Or actually do these manipulations cause similar mitochondrial morphological defects in INPs/GMCs, which in turn impact their proliferation/differentiation? The authors should examine the behavior of type II NBs and mitochondrial morphology in INPs/GMCS after Opa1/Marf/Drp1 knockdown/inhibition or Notch knockdown/overexpression.

5. In terms of mitochondrial morphological defects described in the manuscript, only thing obvious and convincing to me is that expressing Drp1SD leads to aggression of mitochondria. I am not sure if anyone can see the mitochondrial morphological defects described for Opt1/Marf/Notch or overexpressing NotchFL/Nintra without looking at the quantification data. Furthermore, in Drp1SD expressing NBs, mitochondria all aggregate together, how could they measure the area of individual mitochondria. If they measured the region where they could see individual mitochondria as they mentioned, then obviously the data are not representative and does not truly reflect the phenotype observed in Drp1SD expressing NBs.

6. As I mentioned previously, the data presented in this manuscript are not consistent with a published paper (Bonnay F, et al, Cell 2020, 182, 1490) in terms of the number of mINPs in Marf and Opa1 knockdown type II NB lineages. The authors should test the same VDRC RNAi lines used by Bonnay F et all to see if they can reproduce the results. They should also discuss the discrepancy of the results in the discussion part.

7. The authors suggest that Notch signaling regulates mitochondrial fission/fusion by activating the expression of Opa1, which in turn regulates Notch signaling by promoting translocation of NICD from the cytoplasm to the nucleus. So there is a positive feedback loop of Notch signaling and Opa1. However, I don’t find that their data are convincing enough to support that Opa1 regulates Notch signaling by regulating the nuclear import of NICD. The biggest concern I have is that E(Spl)-GFP [by the way, it should read E(Spl)m�-GFP as there are multiple different E(Spl) genes] signal is reduced in type II NBs when prospoero-GAL4 drives the expression of UAS-opa1 RNAi (Fig6E-G). Prospero-GAL4 is not expressed in type II NBs at all. It is only expressed in INPs and GMCs. Therefore, Opa1 is not knocked down in type II NBs at all when UAS-opa1 RNAi is driven by prospero-GAL4. How could E(Spl)m� -GFP expression in type II NBs be affected in this case? By the way, comparing the E(Spl)m�-GFP expression levels by comparing the signal/noise ratio certainly is not the best way for quantifying the E(Spl)m�-GFP expression. Furthermore, I am not totally convinced that NICD staining could be used to assess the activation of Notch receptors in this case. The NICD antibody recognize both the uncleaved Notch as well as cleaved NICD. And I do not see clear NICD signal in the nucleus. In fact, in their discussion part, the authors state that “our observations show that Opa1 depleted type II NBs show a loss of Notch receptors on the plasma membrane and increase in endosomes in the type II NB and in the cells of the lineage”, which is not shown in their figures at all. In Fig 6C-D, they only compared the ratio of cyto/nuclear of NICD staining intensity. Therefore, more convincing data are needed to further support that Opa1 regulates Notch signaling. It is hard to imagine that mitochondria proteins like Opa1 would regulates the translocation of NICD from the cytoplasm to the nucleus. The authors did not provide any reasonable explanation either in the discussion part. If they cannot provide more convincing data, the model proposed in Fig 8 should also be modified.

8. In lines 375-277, the authored stated that “We also observed a change in the mitochondrial morphology …… (Figures S8A-B)”. However, in that figure, they only show the stainining of Dpn and Ase and wild type and NotchFL overexpressing lineages. Also, in this figure, I would expect to see ectopic type II Nbs in NotchFL overexpressing lineages, but the lineages shown in the figure look pretty normal and did not have any extra type II NB lineages. Why?

9. In my previous comments (#12), I suggested to examine the expression of Ase in in Drp1SD Notch RNAi type II NBs to further elucidate how Drp1SD expression could rescue the loss of mINPs in Notch RNAi lineages. However, they still did not do it. In response to my comment, they stated that “2. Ase is not present in the type II neuroblast lineages at the third instar larval stage in Drp1SD; notch RNAi” and cited again Figure S8. However, Fig S8 is about NotchFL overexpression, not notch RNAi. So they cited the same figure wrongly at least twice (see above #8). They also argued that “3. We have Dpn and Ase antibodies in the same animal and hense we could not check for loss of conversion of type I by dissecting brains at earlier time points”. This does not sound a reasonable excuse for not examining the Ase expression in Drp1SD, notch RNAi type II NBs. First, the check the expression of Ase in Drp1SD, notch RNAi type II NBs, it does not require co-staining of Dpn and Ase. Ase staining alone is sufficient. mCD8-GFP driven by pnt-GAL4 would allow for identification of type II NBs. Second, the ectopic Ase expression (thus the conversion of type II NBs into type I NBs can be observed in late larval stages when Notch is knocked down according to published papers.

10. Related with the rescue of Notch RNAi phenotypes by Drp1SD, the authors showed that Drp1SD could rescued the number of mINPs and Ase+ cells but not Pros[high] cells. That does not seem to make sense. The authors should explain how this could happen.

Other minor concerns:

11. In lines 419 – 421, they stated that “the loss of Dpn positive cells in notch RNAi… was suppressed completely … (Fig 7A, E)”, but the bar graph in Fig7E still show statistically significant difference between +/+ and Drp1SD; notch RNAi. So the suppression is not complete. The statement should be modified.

12. In lines 542 – 545, the authors stated that “Drp1 mutant …. that Drp1 is dispensable for NB formation….”. It is an overstatement because the expression of Drp1SD is driven by pnt-GAL4, which is only expressed after type II NBs are formed. Therefore, they cannot conclude that Dpr1 is dispensable for NB formation.

13. Line 627: dilution of the Ase antibody should be indicated.

14. For estimating the number of Ase+ neurons, they used the average area of Pros+ nuclei of 5 GMCs (lines 759-765). However, the size of GMCs may not necessary be the same as neurons. They should use the average area of Pros+ nuclei of neurons instead of GMCs.

15. In fig6C opa1 RNAi, what do the yellow arrows point to? It is not mentioned in the figure legends.

16. Fig 7E, Y-axis, Dpn+ve should be changed to Dpn+

Reviewer #2: The authors have addressed some of my concerns from the previous version of the ms, and in general, the data quality has improved. I however still have several reservations.

I have still a problem with the data shown in Figure 1, the authors claim that there is fragmentation of mitochondria, and indeed in the particular ROI they have chosen, the quantification is reflective of their observation. However, it is not clear to me how this box is chosen in the top three panels of Figure 1B. To make this point more convincing, the authors should choose multiple ROIs in the same section and perform the analysis, as a minimum. Similar point applies to Figure 3A, the claim that the phenotype is to do with fragmentation of the mitochondria is still not very convincing. In Figure 3, seems like the more striking and perhaps easier to quantify phenotype is the aggregation of the mitochondria to one side of the neuroblast, and maybe the total amount of mitochondria is increased in Drp1SD and similarly in combination with opaRNAi and MarfRNAi. Is there an explanation as to why the fusion causes aggregation of the mitochondria to one side of the NB?

In Figure 2H, as the authors are unable to show Dpn and Ase together due to technical reasons, it is a claim too far to say that it is definitely mature INP to GMC transition that is impacted in MarfRNAi. Furthermore, in the summary diagram, if Dpn/Ase double positive cells could not be ascertained, I don’t see how it can be summarised as such in the diagram.

Reviewer #3: The authors have put in a commendable effort to improve the manuscript. They have considered many of the reviewer’s comments and implemented most in some way. In particular, changes to the manuscript writing, figure legends and explanations of methodology and stocks has improved understanding of their presented results. The additional results obtained showing reduced Notch signaling upon Opa1 knockdown are very interesting and provides additional evidence for a link between Notch signaling and mitochondrial morphology in differentiation that will be exciting to follow up.

However, the authors have still failed to convince me about the way that they quantify mitochondrial morphology and distinguish NB lineages/cells. The mitochondrial images presented in Figure 1B are clearer in the revised manuscript, but I doubt whether they are sufficient to allow accurate quantifications of mitochondrial morphology. On top of that, the quantifications were performed on regions manually chosen for those where mitochondria are most distinct, which does not necessary reflect the overall morphology in a given cell. I empathize with the authors as mitochondrial morphology quantification is challenging, but this is a very important part of the manuscript and the current data raise more doubts than adding any strength to the author’s conclusions. Similarly, it is not entirely convincing that the authors were able to distinguish different neuroblast lineages as nearly all quantifications in Figure 2 rely on a single marker, except for Figure 2D for which representative images are unfortunately omitted. This deficiency, plus my concern in the mitochondrial quantification method, makes it hard to judge the reliability of the overall conclusion of the paper. I, therefore, cannot recommend this paper for publication at this stage.

**Have all data underlying the figures and results presented in the manuscript been provided?**

Reviewer #1: Yes

Reviewer #2: Yes

Reviewer #3: Yes

PLOS authors have the option to publish the peer review history of their article (what does this mean?). If published, this will include your full peer review and any attached files.

Reviewer #1: No

Reviewer #2: No

Reviewer #3: No

---

## [Decision Letter · Decision Letter 2]

12 Jan 2022

Dear Dr Rikhy,

Thank you very much for submitting your Research Article entitled 'Mitochondrial fusion regulates proliferation and differentiation in the type II neuroblast lineage in Drosophila' to PLOS Genetics.

The manuscript was fully evaluated at the editorial level and by independent peer reviewers. The reviewers appreciated the attention to an important topic but identified some concerns that we ask you address in a revised manuscript. While we appreciate Reviewer #3’s concern that image quality needs to support the underlying analysis and conclusions, the editors and the other two reviewers are satisfied, so we will not require additional experimentation to acquire new images (though you are welcome to include improved images in the revision if you are able). Please respond to the remaining concerns in your revision.<o:p></o:p>

We therefore ask you to modify the manuscript according to the review recommendations. Your revisions should address the specific points made by each reviewer.

[LINK]

Yours sincerely,

Hongyan Wang, Ph.D.

Associate Editor

PLOS Genetics

Gregory P. Copenhaver

Editor-in-Chief

PLOS Genetics

Reviewer's Responses to Questions

**Comments to the Authors:**

Reviewer #1: I appreciate that the authors put a tremendous amount of effort into the revision. The manuscript has been improved significantly and most of my concerns have been addressed. Although I still have some reservation about the quantification of the area of mitochondria, the additional quantification of clustered/dispersed/fragmented mitochondria is helpful to some extent. I only have two suggestions.

First, If the authors insist on quantifying the number of neurons by labeling the type II NB lineages with mCD8-GFP driven by pntP1-GAL4 and Pros staining, they should state clearly that they are quantifying only young neurons, not the total number of neurons. Strong Pros expression is only in young/newly-born neurons, not in old neurons. Furthermore, the expression mCD8-GFP driven by pntP1-GAL4 certainly does not label all the cells in type II NB lineages (as they also stated in the response letter that sometimes they could not determine the boundary of the lineage based on the mCD8-GFP expression).

Second, I would suggest to provide bar graphs for the quantification of clustered/dispersed/fragmented mitochondria in the figures (1, 3, 6, S10) instead of providing the numerical data in the figure legends. Also, they should clearly describe these quantification data in the text of the Results part.

Reviewer #2: The revisions were done to my satisfaction.

Reviewer #3: Although the morphological changes of the mitochondrial network are better described in the revised version, the poor quality of microscope images in multiple figures still failed to convince us that they are sufficient to allow reliable quantifications of mitochondrial morphology. We, therefore, would like to keep our decision from the previous round of revision.

**Have all data underlying the figures and results presented in the manuscript been provided?**

Reviewer #1: Yes

Reviewer #2: Yes

Reviewer #3: Yes

PLOS authors have the option to publish the peer review history of their article (what does this mean?). If published, this will include your full peer review and any attached files.

Reviewer #1: No

Reviewer #2: No

Reviewer #3: No

---

## [Decision Letter · Decision Letter 3]

27 Jan 2022

Dear Dr Rikhy,

We are pleased to inform you that your manuscript entitled "Mitochondrial fusion regulates proliferation and differentiation in the type II neuroblast lineage in Drosophila" has been editorially accepted for publication in PLOS Genetics. Congratulations!

Please note that Reviewer #1 (see below) has indicated that an issue with the scale bar in several figures needs attention.  Please address this as you prepare your final draft for the production team (the editorial team will not need to re-evaluate).

Yours sincerely,

Hongyan Wang, Ph.D.

Associate Editor

PLOS Genetics

Gregory P. Copenhaver

Editor-in-Chief

PLOS Genetics

Comments from the reviewers (if applicable):

Reviewer's Responses to Questions

**Comments to the Authors:**

Reviewer #1: The authors have adequately addressed my concerns.

Only one minor thing about scale bars. They need to state clearly which scale bar represents 5um and which one 2.7um in the figure legends for Fig 1 and Fig 3. What scale bars represent in Figs S5, S6, S7 S8, and S9 should also be stated in their figure legends.

**Have all data underlying the figures and results presented in the manuscript been provided?**

Reviewer #1: Yes

PLOS authors have the option to publish the peer review history of their article (what does this mean?). If published, this will include your full peer review and any attached files.

Reviewer #1: No

**Data Deposition**

http://datadryad.org/submit?journalID=pgenetics&manu=PGENETICS-D-20-01908R3

**Press Queries**

---

## [Editor Report · Acceptance letter]

8 Feb 2022

PGENETICS-D-20-01908R3 

Mitochondrial fusion regulates proliferation and differentiation in the type II neuroblast lineage in Drosophila 

Dear Dr Rikhy, 

We are pleased to inform you that your manuscript entitled "Mitochondrial fusion regulates proliferation and differentiation in the type II neuroblast lineage in Drosophila" has been formally accepted for publication in PLOS Genetics! Your manuscript is now with our production department and you will be notified of the publication date in due course.

With kind regards,

Katalin Szabo

PLOS Genetics

On behalf of:
